# Learning to Delegate for Large-scale Vehicle Routing

**Sirui Li**[*]
MIT
siruil@mit.edu

**Zhongxia Yan**[*]
MIT
zxyan@mit.edu

**Cathy Wu**
MIT
cathywu@mit.edu

## Abstract

Vehicle routing problems (VRPs) form a class of combinatorial problems with wide practical applications. While previous heuristic or learning-based works achieve decent solutions on small problem instances, their performance deteriorates in large problems. This article presents a novel learning-augmented local search framework to solve large-scale VRP. The method iteratively improves the solution by identifying appropriate subproblems and *delegating* their improvement to a black box subsolver. At each step, we leverage spatial locality to consider only a linear number of subproblems, rather than exponential. We frame subproblem selection as regression and train a Transformer on a generated training set of problem instances. Our method accelerates state-of-the-art VRP solvers by 10x to 100x while achieving competitive solution qualities for VRPs with sizes ranging from 500 to 3000. Learned subproblem selection offers a 1.5x to 2x speedup over heuristic or random selection. Our results generalize to a variety of VRP distributions, variants, and solvers.

## 1 Introduction

Vehicle routing problems (VRPs) have enjoyed ample applications in logistics and ride-hailing services [23] around the world for decades. While determining the optimal solution to a VRP is NP-hard [25], there have been numerous attempts to solve VRPs both exactly and approximately: provable algorithms have been designed for specific problem instances up to size 130 [28], and powerful heuristic solvers such as LKH-3 [13] and HGS [49, 48] find good solutions in practice for problems of size more than 1000. However, heuristics methods often suffer from inflexibility due to extensive hand-crafting and the heavy computational burden from lengthy iterative procedures, as in the case of LKH-3. For example, LKH-3 takes more than an hour to solve a size 2000 CVRP instance (Table 1), which is impractical for applications such as large-scale courier or municipal services.

More recently, machine learning methods inspired by the Pointer Network [50] provide alternatives to traditional solvers: the learning-based methods greatly reduce computation time while maintaining decent solution quality on small problem instances (less than 100 cities), by training on diverse sets of problem distributions either via supervised [50] or reinforcement learning [29, 20]. However, these methods remain difficult to scale, and few report results on problems of size more than 200.

Our work aims to address scalability by learning to identify smaller subproblems which can be readily solved by existing methods. Our learned subproblem selector guides the problem-solving process to focus local improvement on promising subregions. While there exists a combinatorial number

---

[*]Equal Contribution

35th Conference on Neural Information Processing Systems (NeurIPS 2021).

of subproblems that can be selected at each step, we leverage spatial locality commonly found in combinatorial optimization problems to restrict the selection space size to be linear in the problem size. Intuitively, objects far away from each other generally have very small influence on each other's solution and are likely to be in different routes, so they should not be part of the same subproblem. The greatly reduced search space enables us to feasibly train an attention-based subproblem selector.

Our framework combines the advantages of learning and heuristics: our network identifies promising subproblems to improve upon, dramatically speeding up solution times. Using a competitive subsolver on subproblems, we achieve good solution quality without the high computational costs of running the subsolver on large problem instances. In summary, our contributions are:

- We propose *learning-to-delegate*, a learning-based framework for solving large-scale VRPs by iteratively identifying and solving smaller subproblems.
- Despite the high dimensionality and NP-hardness of subproblems, we design a Transformer architecture that effectively predicts the subsolver's solution quality for a subproblem.
- With extensive validation, we show that learning-to-delegate offers significant speedups and/or objective improvements over both its base subsolver and random (or heuristic) subproblem selection, for a variety of VRP variants, distributions, and solvers.

## 2 Preliminary: Capacitated Vehicle Routing Problems (CVRP)

In CVRP, there is a depot node 0 and city nodes $\{1, ..., N\}$. Each city node $i$ has demand $d_i$ to fulfill. A vehicle with capacity $C$ starts and ends at the depot and visits a route of city nodes such that the sum of city demands along the route does not exceed $C$, after which the vehicle starts a new route again. The objective is to find a valid solution minimizing the solution cost. We define the following:

- Route: a sequence of nodes, where the first and last node are the depot 0, and the rest are city nodes. In a valid route, the sum of demands of the city nodes does not exceed $C$.
- Route cost: the sum of edge costs for the sequence of nodes. For an edge from node $i$ to node $j$, the edge cost is the Euclidean distance between node $i$ and $j$.
- Solution: a *feasible* solution consists of a set of valid routes visiting each city exactly once.
- Solution cost: the sum of route costs for all routes in the solution. An *optimal* solution is a feasible solution with the minimum solution cost.
- Subproblem: a CVRP consisting of the depot, a subset of cities, and corresponding demands.

## 3 Related Work

**Classical methods.** Heuristics for solving combinatorial optimization problems have been studied for decades. The most powerful methods, such as local search [1], genetic algorithms [36], and ant colony methods [10], involve iteratively improving the solution in a hand-designed neighborhood. For example, move, swap [52], and 2-opt [9] are well-known heuristics for the traveling salesman and vehicle routing problems. The competitive VRP solver LKH-3 [13] uses the Lin–Kernighan heuristic [26] as a backbone, which involves swapping pairs of sub-routes to create new routes, whereas the CVRP solver HGS [49, 48] uses a hybrid genetic and local search procedure to achieve state-of-the-art solution qualities on problems up to size 1000. While LKH-3[2] tackles a variety of VRP variants, the publicly available implementation of HGS [3] only solves CVRP.

For large problems, low-level heuristics are combined with *meta-heuristics*, including Tabu Search with Adaptive Memory [40], guided local search [51], and Large Neighborhood Search [35]. The inspiration for our work derives from Partial OPtimization Metaheuristic Under Special Intensification Conditions (POPMUSIC), which iteratively optimizes problem subparts and has been used to solve problems as diverse as map labeling [24], berth allocation [22], and $p$-median clustering [39].

Despite the promise of the POPMUSIC framework for large-scale combinatorial optimization, the impact of certain design choices is not well understood, such as the subproblem selection ordering

---

[2]We refer to the LKH-3 code at `http://webhotel4.ruc.dk/~keld/research/LKH-3/`.
[3]We refer to the HGS code at `https://github.com/vidalt/HGS-CVRP`.

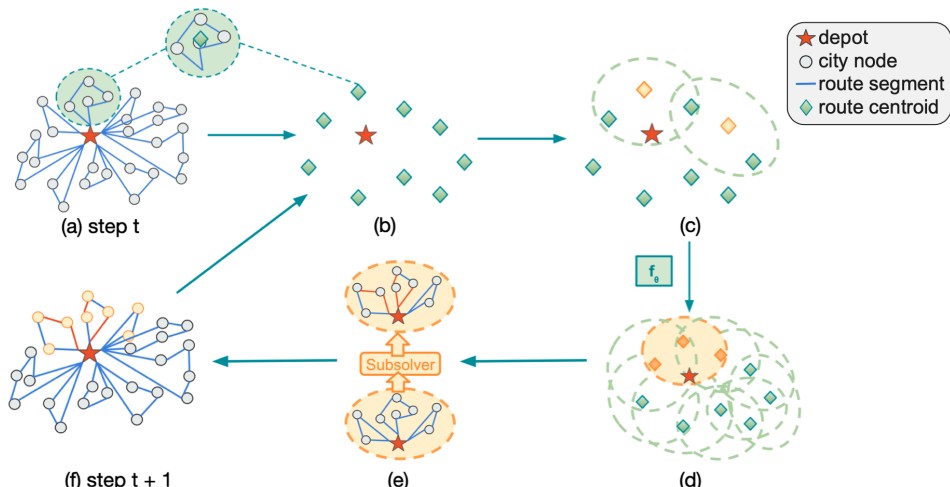

Figure 1: **Our iterative framework for VRPs**. (a) At each time step, we start with a current solution $X$. Circles are city nodes, blue lines are route segments, and the red star is the depot. (b) We aggregate each route by taking the centroid of all city nodes in the route. (c) For each route, we define a corresponding subproblem as the $k$-nearest neighbors of the route. Two such subproblems with $k = 3$ are shown. (d) Our subproblem selector selects a subproblem $S$ (yellow). (e) We feed $S$ into the subsolver to get a new subsolution $X'_S$. The red edges are updated by the subsolver. (f) We update $X$ to new solution $X'$ with $X'_S$, then repeat (b)-(f).

and size [43]. For example, it is not clear how to meaningfully order the subproblems. One early work in this direction demonstrated that a last-in-first-out stack order performs better than random [3]. Our work provides a natural approach to order the subproblems based on predicted improvement.

**Deep learning methods.** Recently, there has been a surge of interest in training deep neural networks (DNN) to solve combinatorial problems. As observed by Kwon et al. [21], most methods fall into one of the following two categories: 1) *construction methods* [29, 20], where an autoregressive model such as the Pointer Network [50] directly outputs a solution, and 2) *improvement methods* [7, 15], where the DNN iteratively performs local updates to the solution, resembling local search.

These methods are approaching the solution quality of LKH-3 [13] on small problem instances. For example, Kwon et al. [21] extends a construction approach [20] to encourage more diverse output samples on $N \leq 100$ VRPs. Among improvement methods, Lu et al. [27] learn a meta-controller to select among a set of local search heuristics, marking the first learning-based approach to outperform LKH-3 on $N \leq 100$ VRPs. Despite these successes, learning-based approaches for large-scale VRPs are poorly understood. In addition, all of the aforementioned methods are trained using deep reinforcement learning; for large problems, trajectory collection becomes prohibitively expensive.

**Scaling up learning methods.** A few recent works have begun to investigate scaling of learned networks for NP-hard graph problems. For example, Ahn et al. [2] propose an iterative scheme called *learning-what-to-defer* for maximum independent set. At each stage, for each node in the graph, the network either outputs the solution for the node or defers the determination to later stages. Song et al. [38] proposes an imitation-learning-based pipeline for Integer Linear Programs (ILP), where at each stage they partition all variables into disjoint subsets, and use the Gurobi [12] ILP solver to solve the partitions. Due to differences in graph structure, our work presents a more natural scheme to handle VRP constraints and structures. A few works attempt to incorporate learning to decompose large-scale VRPs [6, 47, 31]. However, the decomposition approaches proposed appear to be experimentally less effective than ours.

## 4 An Iterative Framework for VRPs

While CVRPs are NP-hard and thus require worst-case exponential time in the problem size to solve optimally, typical CVRPs exhibit structure that practical solvers may exploit. We hypothesize that in

such situations, the larger problem can be efficiently approximately solved as a sequence of smaller subproblems, which can be *delegated* to efficient subsolvers. To test this hypothesis, we propose a learning-based, iterative framework with two components: 1) a *subsolver* capable of solving a small problem instance (exactly or approximately), and 2) a *learned model* for identifying suitable subproblems within a larger problem to delegate to the subsolver.

---

**Algorithm 1:** Learning to Delegate

---

**Input:** Problem instance $P$, initialized solution $X$, subproblem selector $f_\theta$, subsolver
$\quad\quad\quad Subsolver$, number of steps $T$, parameter $k$ denoting the size of subproblems

1 **for** *Step $t$ = 1: $T$* **do**
2 $\quad$ $\mathcal{S}_{k,\text{local}} \leftarrow \text{ConstructSubproblems}(P, X, k)$
3 $\quad$ $S \leftarrow f_\theta(\mathcal{S}_{k,\text{local}})$
4 $\quad$ $X'_S \leftarrow \text{Subsolver}(S)$
5 $\quad$ $X \leftarrow X'_S \cup X_{P \setminus S}$
6 **end for**

---

We illustrate our iterative framework in Figure 1, which takes in a problem instance $P$ with a feasible initial solution $X_0$. At each step, given the current solution $X$, we select a smaller subproblem $S \subset P$ with our learned model $f_\theta$ then apply the subsolver to solve $S$; we then update $X$ with the new solution for $S$. To maintain feasibility after an update, we restrict $S$ to be the set $\mathcal{S}$ of visited cities of a *subset of routes* from $X$. Since routes with cities in $P \setminus S$ remain valid routes, we obtain a new feasible solution $X' = X'_S \cup X_{P \setminus S}$, where $X'_S$ is the subsolution for $S$ and $X_{P \setminus S}$ consists of unselected routes from $X$. Intuitively, a strong $f_\theta$ should identify a subproblem $S$ such that the subsolver solution $X'_S$ results in a large improvement in objective from $X$ to $X'$.

### 4.1 The Restricted Subproblem Selection Space

As the number of routes in $P$ is $R = O(N)$, the cardinality of the selection space $\mathcal{S}$ is $O(2^R)$, which is exponential in the problem size $N$ and difficult for learned models to consider. If we restrict each subproblem to cities from exactly $k$ routes, there are still $\binom{R}{k} = O(R^k)$ subproblems to consider. Therefore we further restrict selection to subproblems with spatial locality. As shown in Figure 1(d), we only consider subproblems from $\mathcal{S}_{k,\text{local}}$, where each subproblem is centered around a particular route $r$ and contains the $k$ routes whose centroids have the smallest Euclidean distance to the centroid of $r$. In this way, we reduce the selection space to $|\mathcal{S}_{k,\text{local}}| = R = O(N)$ from $|\mathcal{S}| = O(2^N)$. In Algorithm 1, we refer to this restriction as ConstructSubproblems. Our restriction to a local selection space is motivated by the fact that many combinatorial optimizations problems have inherent spatial locality, i.e. problem entities are more strongly affected by nearby entities than faraway entities. Earlier heuristical methods such as POPMUSIC [42] leverage similar spatial locality.

## 5 Learning to Delegate

In this section, we discuss criteria for selecting subproblems, and how to train the subproblem selector.

**Improvement as the criteria for subproblem selection.** Given a selected subproblem $S$ with a current solution $X_S$ on the subproblem, we obtain from LKH-3 a new solution $X'_S$ on the same subproblem (step $(d)$ to $(e)$ in Figure 1). We then define the *immediate improvement*

$$\delta(S) = c(X_S) - c(X'_S) \tag{1}$$

where $c(X_S)$ is the total cost of subsolution $X_S$ and $\delta(S)$ is the improvement in the solution cost. In this way, the sum of improvements along $T$ steps is the total improvement in solution quality. As we empirically find that providing the previous subsolution $X_S$ to the subsolver may trap the subsolver in a local optimum, we withhold $X_S$ from the solver and thus may see non-positive improvement $\delta(S) \leq 0$, especially after many steps of subproblem selection. At test time, to avoid worsening the objective, we adopt a hill-climbing procedure such that when $\delta(S) \leq 0$, we keep $X_S$ instead of $X'_S$ (Figure 1, step $(e)$). With proper masking, we avoid selecting the same non-improving subproblem $S$ again. The hill-climbing and masking procedures are applied to both our subproblem-selection network $f_\theta$ and the three heuristic selection rules, described in 6.1, to maintain fair comparisons.

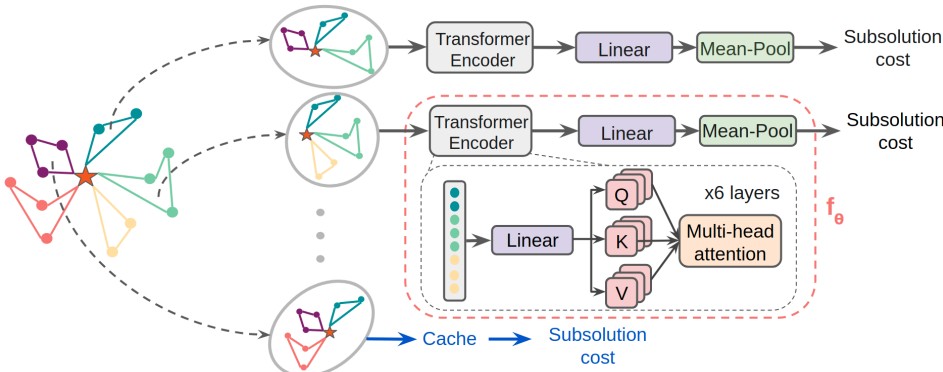

Figure 2: **Our Transformer architecture**. At each step of our framework (Figure 1(c)), we featurize subproblem $S \in \mathcal{S}_{k,\text{local}}$ into a column vector of unordered cities. We apply the Transformer encoder with multiple multi-head attention layers and a final linear layer, before mean-pooling over the cities to generate the predictions $f_\theta(S)$, which we fit to the subsolution cost $c(X'_S)$. When possible, we retrieve a previously predicted subproblem from a cache to minimize computation.

**Subproblem selection.** The goal of our subproblem selector is to select the subproblem leading to the best *immediate* improvement. While our subproblem selection does not directly optimize for the total improvement, we observe that (1) our subsolver may perform many low-level operations internally, so high-level problem selection may still benefit greatly from maximizing the immediate improvement, and (2) numerous reinforcement learning approaches to VRP [53, 18] choose a small discount factor such as $\gamma = 0.25$ when optimizing a multi-step objective $\sum_{t=1}^{T} \gamma^t \delta_t$, as doing so encourage faster convergence. Moreover, our subproblem selector may instead be trained on multi-step search data to select subproblems offering the best multi-step improvement.

**Ground-truth labels.** Our restricted selection space allows us to enumerate all possible subproblems at each step. In a typical large scale CVRP instance with $N = 2000$ cities, a solution consists of roughly $R = 200$ routes, so the size of the selection space is 200. We obtain the immediate improvement $\delta(S)$ by running the subsolver on each subproblem $S \in \mathcal{S}_{k,\text{local}}$. Although our enumeration strategy is feasible for generating training data, it is much too slow to execute on a test CVRP instance. However, if our subproblem selector can predict the best immediate improvement at test time (that is, without running the subsolver on multiple subproblems), then we can combine the best of both worlds to obtain a fast and accurate selection strategy.

**Selection strategies: regression vs classification.** Given a labeled dataset, we may treat the task of identifying the best subproblem as either regression or classification. In the context of imitation learning with a greedy expert, the former learns a *value function* while the latter learns a *policy*. Due to space limitation, we focus discussion on regression and reserve comparison with classification for Appendix A.5.1. The regression-based subproblem selector uses a trained $f_\theta$ to predict the subsolution cost $c(X'_S)$; we then simply compute $\arg\max_S c(X_S) - f_\theta(S)$ to select the best subproblem.

**Network architecture.** We define $f_\theta$ with a Transformer encoder architecture [45]. The input representing each subproblem $S$ is the unordered set of featurized cities in $S$. The features for each city consist of the demand $d$ and the location of the city $(x, y)$ relative to the depot. As we do not feed the existing subsolution $X_S$ to the subsolver, the dense multi-head attention mechanism does not need to be modified to take the routes in $X_S$ into account. The output of the Transformer encoder is fed into a linear layer then mean-pooled to obtain the scalar prediction $f_\theta(S)$. In Appendix A.5.5, we perform an ablation study with simpler architectures.

**Loss function.** We empirically find mean squared error to be less stable than Huber loss [16], which we set as our loss function

$$L(\theta; S) = \begin{cases} \frac{1}{2}(f_\theta(S) - c(X'_S))^2, & \text{if } |f_\theta(S) - c(X'_S)| \leq 1 \\ |f_\theta(S) - c(X'_S)| - \frac{1}{2}, & \text{otherwise} \end{cases} \tag{2}$$

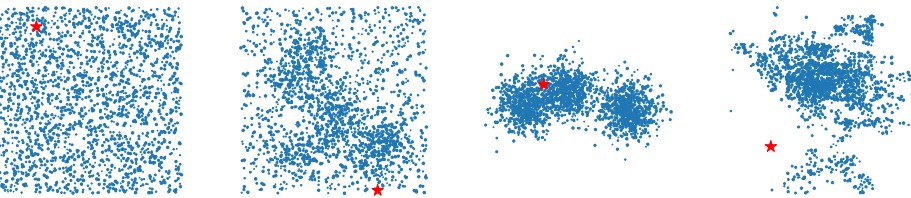

Figure 3: **Instances from** $N = 2000$ **CVRP distributions**. From left to right: instance from uniform, mixed ($n_c = 7$ cluster centers), clustered ($n_c = 3$ cluster centers), and real-world. The red star is the location of the depot, while blue dots are cities sized proportional to demand.

# 6 Experiments and Analysis

We illustrate the CVRP distributions considered in our work in Figure 3. We perform extensive experiments to evaluate our learning framework, aiming to answer the following questions:

1. **Uniform distribution**. How does our method compare with baselines, in terms of solution time and quality, on problems with uniformly distributed cities?

2. **Clustered distributions**. How does our method perform on problems with clustered cities?

3. **Out-of-distribution**. Can our model generalize, such as to larger or real-world instances?

4. **VRP variants**. Can our method address more sophisticated VRPs? E.g., CVRP with Time Windows (CVRPTW) [37] or VRP with Mixed Pickup and Delivery (VRPMPD) [33].

5. **VRP solvers**. Can our method be adapted to leverage other VRP subsolvers?

We reserve additional ablations on subproblem selection as classification, effect of subproblem size $k$ and discussion of asymptotic behavior, subproblem selection with the HGS subsolver, effect of weaker initialization methods, and comparison with simpler architectures for Appendix A.5.

## 6.1 Setup

We briefly describe experimental setup in the main text and defer full details to Appendix A.1. Given a particular distribution of VRP, we generate separate sets of training, validation, and test problem instances. Unless otherwise stated, our validation and test sets contain 40 instances each. For each problem instance, we generate a rough initial solution by partitioning it into disjoint subsets of cities and briefly running the subsolver on each subset. Due to its compatibility with many VRP variants, we use LKH-3 [13] as the subsolver for all VRP distributions unless otherwise stated.

We generate the training set by running our iterative framework and selecting subproblems by enumerating the subsolver on all $S \in \mathcal{S}_{k,\text{local}}$. As many subproblems remain unchanged from $X$ to $X'$, we use previously cached subsolutions when possible instead of re-running the subsolver. While generation times differ depending on several factors, typically it takes less than 10 hours with 200 Intel Xeon Platinum 8260 CPUs to generate a training set of 2000 instances.

To avoid training multiple models for different problem sizes $N$, we train a single model for each VRP distribution with combined data from multiple $N \in \{500, 1000, 2000\}$. Training takes at most 8 hours on a single NVIDIA V100 GPU. To exploit symmetry in problem instances, we apply rotation and flip augmentation at training time. To evaluate trained selectors on a problem instance, we run the iterative selection framework on a single CPU, with disabled multithreading and no GPU.

We select the best hyperparameters via manual search based on validation set performance. We record final results on the test set as the mean and standard error over 5 runs with different random seeds.

**Baselines.** By default, as we use LKH-3 as the subsolver, we run LKH-3 on the full problem instances with our initialization for 30000 local update steps, which takes 2-3 hours on average for $N = 2000$. If using HGS as the subsolver, we also run HGS on the full problem instances for the same amount of time as our LKH-3 baseline. These baselines allow us to compute the speedup of our framework over the subsolver alone. We also compare against OR Tools [30], another open source

Table 1: **Performance and computation time for uniform CVRP**. For problem instance sizes $N \in \{500, 1000, 2000\}$, we report the objective values (*lower* is better) of our method and baseline methods, averaged across all instances in the test set. Note that the cost is the total distance of routes in the solution. LKH-3 (30k) runs LKH-3 for 30k steps to near convergence, while LKH-3 (95%) is the 95% solution quality. Random, Count-based, Max Min Distance, and Ours (Short) run until matching LKH-3 (95%) in solution quality, with the speedup reported in parentheses, while Ours (long) runs for twice amount time as Ours (Short).

| Method | N = 500 | | N = 1000 | | N = 2000 | |
| --- | --- | --- | --- | --- | --- | --- |
| | Cost | Time | Cost | Time | Cost | Time |
| LKH-3 (95%) | 62.00 | 4.4min | 120.02 | 18min | 234.89 | 52min |
| LKH-3 (30k) | 61.87 | 30min | 119.88 | 77min | 234.65 | 149min |
| OR Tools | 65.59 | 15min | 126.52 | 15min | 244.65 | 15min |
| AM sampling | 69.08 | 4.70s | 151.01 | 17.40s | 356.69 | 32.29s |
| AM greedy | 68.58 | 25ms | 142.84 | 56ms | 307.86 | 147ms |
| NeuRewriter | 73.60 | 58s | 136.29 | 2.3min | 257.61 | 8.1min |
| Random | 61.99 | 71s (3.8x) | 120.02 | 3.2min (5.5x) | 234.88 | 6.4min (8.0x) |
| Count-based | 61.99 | 59s (4.5x) | 120.02 | 2.1min (8.2x) | 234.88 | 5.3min (10x) |
| Max Min | 61.99 | 59s (4.5x) | 120.02 | 2.5min (7.0x) | 234.89 | 5.2min (10x) |
| Ours (Short) | 61.99 | 38s (7.0x) | 119.87 | 1.5min (12x) | 234.89 | 3.4min (15x) |
| Ours (Long) | **61.70** | 76s | **119.55** | 3.0min | **233.86** | 6.8min |

heuristic VRP solver employing iterative search, terminating runs at 15 minutes, as OR Tools stops improving the solution within this time for all instances.

We include results for previous learning methods AM [20] and NeuRewriter [7]. These do not outperform LKH-3 even on small problem sizes, and learning methods have had more difficulty generalizing to larger instances. In fact, these methods are trained on problems of size $N \leq 100$, and we find that they yield poor solutions on $N \geq 500$ without architecture modifications and extensive re-training. The AM and NeuRewriter results demonstrate the difficulty of scaling up previous learning methods. We do not initialize OR-Tools, AM, and NeuRewriter because we empirically find that these methods have limited solution capability and do not improve our decent initialization.

To validate our subproblem selector's ability to identify promising subproblems, we design three additional baselines that employ our iterative framework using *hand-crafted heuristics* to select subproblems. The three heuristics that we use are: (1) Random, selecting subproblem $S$ from $\mathcal{S}_{k,\text{local}}$ uniformly; (2) Count-based, which avoids repetitive selections by selecting the subproblem centered at the route whose city nodes have been selected cumulatively the least often in previous steps; and (3) Max Min Distance, which encourages coverage of the entire problem instance by selecting subproblems with the maximum distance from the nearest centroid of previously selected subproblems. We run the heuristic baselines with the same setup as our learned subproblem selector.

**Metrics.** We refer to two metrics to compare our method against baseline methods.

1. **Improvement over method X**: at a specified computation time, the improvement of method Y over method X is the total improvement of method Y minus that of method X.

2. **Speedup over method X**: at a specified solution quality that method X attains, the speedup of method Y over method X is the computation time required for method X to attain the solution quality divided by the time for method Y to attain the solution quality.

We define *95% solution quality* of running a method X over a computation time as the solution quality with 95% of the total improvement. We report speedup at 95% solution quality of a method X because X may take a disproportionate amount of time on the last 5% of improvement; reporting speedup at 100% solution quality inflates the speedup.

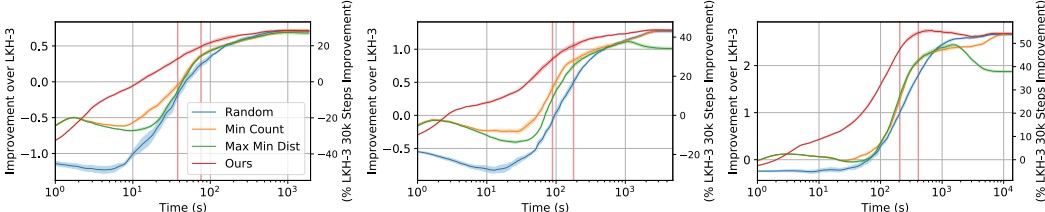

Figure 4: **Improvement over LKH-3 for uniform CVRP**. The x-axis is the computation time and extends until LKH-3 has completed 30k steps. The vertical lines represent the computation times of Ours (Short) and Ours (Long) from Table 1. The three subplots correspond to $N = 500$ (left), $N = 1000$ (middle), and $N = 2000$ (right).

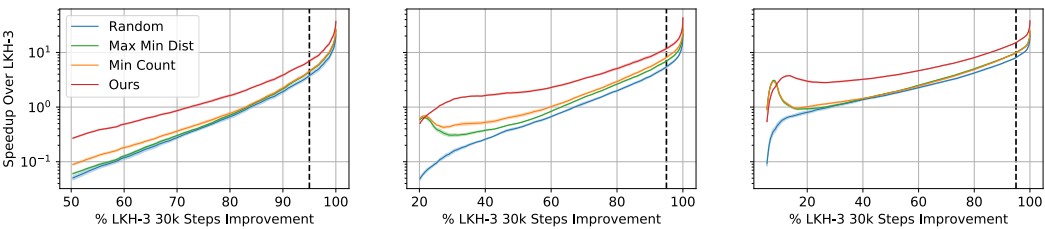

Figure 5: **Speedup over LKH-3 for uniform CVRP**. The x-axis is the solution quality attained, measured as a percentage of the LKH-3's maximum improvement. The dashed vertical line represents the 95% solution quality used to compute the speedup, as mentioned in Table 1. The three subplots correspond to $N = 500$ (left), $N = 1000$ (middle), and $N = 2000$ (right).

## 6.2 Uniform CVRP Distribution

As seen in Table 1, our method achieves the best performance for all problem sizes, matching LKH-3's solution quality with more than 7x to 15x less computation time and offering even more improvements with longer computation time. Although we need to evaluate all $R = O(N)$ subproblems of the initial solution with our subproblem selector, subsequent per-step computation time of our method is mostly independent of the problem size $N$ since we only evaluate changed subproblems.

Running LKH-3 for 30k local update steps achieves superior performance to all previous other heuristic and learning-based baselines. Its solution quality scales well to large problem sizes, yet the solution time is significantly longer. Previous learning-based methods, though fast, result in much worse solution qualities. Our heuristic baselines Random, Count-based, and Max Min Distance demonstrate that our iterative framework, even without the learned subproblem selector, may achieve over 5x to 10x speedup over LKH-3. Nevertheless, our results demonstrate that learning the subproblem selector may offer an additional 1.5x speedup over non-learning heuristics.

In Figure 4, we demonstrate the solution quality of our method and baselines compared to LKH-3. We see that, compared to baseline methods, the learned subproblem selector obtains the best solution quality when run for a reasonable amount of time. The improvement of most methods based on our iterative framework converge when run for an excessive amount of time; this is unsurprising, as Random and other baselines are eventually able to select all subproblems offering improvements.

In Figure 5, we demonstrate the speedup of our method over LKH-3, comparing to baseline methods. The speedup is often significant even at low levels of solution quality, and improves with higher solution quality. The speedup is not as meaningful beyond 95% LKH-3 solution quality, as LKH-3 takes a disproportionate amount of time to attain the last 5% of improvement.

## 6.3 Clustered and Mixed CVRP Distributions

We further examine the framework's performance on CVRP distributions with clusters of cities, such as studied in [8, 44]. We generate a clustered instance by first sampling $(x, y)$ locations of $n_c$ cluster centroids then sampling the cities around the centroids. The mixed distribution samples 50% of the cities from a uniform distribution and the rest from around cluster centers. We generate a dataset

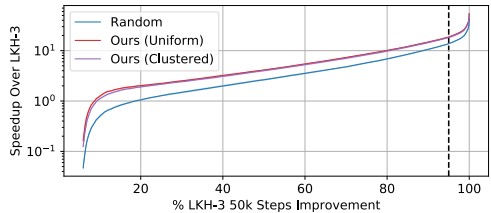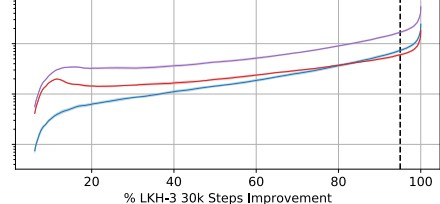

Figure 6: **Speedup in out-of-distribution CVRPs**. The speedup of our model on the $N = 3000$ uniform distribution (left) and $N = 2000$ real-world distribution (right) without finetuning. Ours (Uniform) and Ours (Clustered) are trained on the uniform and clustered CVRP distributions, respectively. Note that LKH-3 is run for 50k steps for $N = 3000$ instances.

of instances for every $(N, n_c, m) \in \{500, 1000, 2000\} \times \{3, 5, 7\} \times \{\text{Clustered}, \text{Mixed}\}$ and train a single model on the entire dataset. We evaluate the model on validation and test sets of 10 instances per $(N, n_c, m)$ combination (i.e. 60 instances per $N$). Due to space limitation, we provide more details about the data distribution and full results in Appendix A.2.

Table 2 reports speedups of our method and the Random baseline over LKH-3. Our method sees at least 2x speedup over Random in all settings. We see larger speedups for clustered distributions than for mixed or uniform distributions (Table 1).

Table 2: **Speedup for** $N = 2000$ **clustered and mixed CVRPs** at 95% LKH-3 30k solution quality.

| Setting | | $n_c = 3$ | $n_c = 5$ | $n_c = 7$ |
|---|---|---|---|---|
| Clustered | Ours | 26x | 18x | 25x |
| | Random | 11x | 7.5x | 9.0x |
| Mixed | Ours | 13x | 14x | 14x |
| | Random | 6.6x | 6.4x | 7.6x |

### 6.4 Out-of-distribution Generalization

We study how our subproblem selector generalize to a uniform CVRP distribution with larger problem size $N = 3000$ and to a real-world CVRP distributions, both unseen at training time. The real-world CVRP distribution derives from a CVRP dataset [4] consisting of 10 very-large-scale instances on 5 Belgium regions with $N$ ranging from 3000 to 30000 and randomly generated demand distribution. To generate an instance, we subsample the cities in a region without replacement to $N = 2000$, while regenerating the demands to match our training distribution. For each original instance, we generate 5 subsampled instances to form the test set of 50 total instances. We visualize the original and subsampled datasets in Appendix A.3.

We apply subproblem selectors trained on uniform and clustered data without finetuning on the new data distributions and report the speedup comparison with the Random baseline in Figure 6. We see that when transferring to $N = 3000$, subproblem selectors trained on uniform and clustered data offer similar performance. However, the model trained on the clustered distribution generalizes well to the real-world distribution while the model trained on the uniform distribution fails to generalize, with worse speedup than the Random baseline. These results suggest that the domain variability of the clustered distribution strongly improves generalization performance.

### 6.5 Other VRP Variants

While our previous experiments vary the distribution of city locations in CVRP, here we consider two VRP variants with uniform city distribution but with additional constraints: CVRPTW [37] and VRPMPD [33]. The former specifies a hard time window constraint for visiting each city, while the latter specifies pick up and delivery constraints in addition to capacity constraint. A detailed description of the variants can be found in Appendix A.4.

Similar to CVRP, we observe significant speedup with our iterative framework alone, while learning offers additional speedup. For CVRPTW, our method offers a 8.2x speedup while our Random baseline offers a 5.9x speedup; for VRPMPD, our method offers a 31x speedup while our Random baseline offers a 20x speedup. We suspect that the time window constraint in CVRPTW imposes strict orderings on the order of city visitations, increasing the difficulty for the subproblem selector.

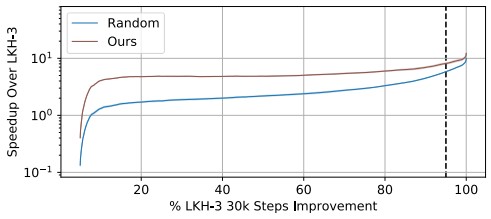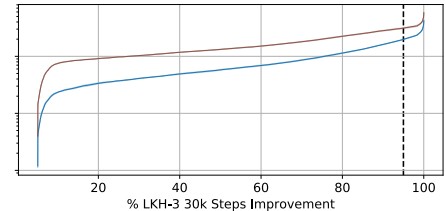

Figure 7: **Speedup in other** $N = 2000$ **VRP variants**, CVRPTW (left) and VRPMPD (right).

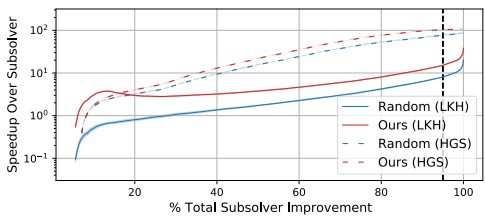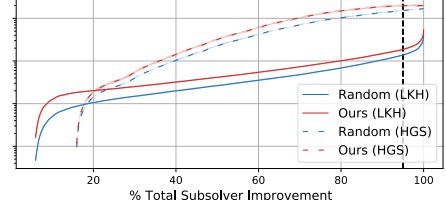

Figure 8: **Speedup with the HGS subsolver on uniform CVRP**. We compare the speedup of our method equipped with LKH-3 or HGS as the subsolver for $N = 2000$ (left) and $N = 3000$ (right).

## 6.6 A State-of-the-Art CVRP Subsolver: HGS

While we focus our analysis on LKH-3 due to its applicability to many variants of VRPs, HGS [49, 48] is a state-of-the-art solver focused on CVRP. Thus, we apply our method to train subproblem selectors for the HGS subsolver. We discuss the full experimental setup and results in Appendix A.5.3.

With HGS as the subsolver, we observe a 103x speedup for our method on $N = 2000$, compared with a 77x speedup for the Random baseline. Similarly, we observe a 198x speedup for our method on $N = 3000$, compared with a 152x speedup for our Random baseline. The large speedup may be due to the fact that HGS is designed and calibrated for medium-scale problems of 500 to 1000 cities, allowing it to function better as a subsolver for large-scale VRPs.

## 7 Conclusion

This paper presents a learning-based framework which learns which subproblems to delegate to a subsolver when solving large VRPs. Spatial locality allows us to learn the subproblem selector over a reduced selection space. The proposed method accelerates competitive VRP solvers on problems of sizes up to 3000, requiring an order of magnitude less computation time. We identify a 1.5x to 2x speedup over non-learning selection strategies. Our results generalize to a variety of VRP distributions, variants, and solvers.

While most previous learning-based combinatorial optimization methods [20, 21, 2] rely on reinforcement learning due to the unavailability of optimal solutions as high-quality labels, our work highlights the counterintuitive effectiveness of supervised learning with only *moderate-quality* labels to achieve high-quality solutions with iterative subproblem selection. An interesting line of future work may explore other ways to effectively leverage moderate-quality labels for combinatorial optimization tasks. In particular, we discuss in Appendix A.6 the applicability of our method to other combinatorial optimization problems with spatial locality. We believe that our learning framework can serve as a powerful technique for both the learning and operations research communities to scale up combinatorial optimization solvers.

Our code is publicly available at `https://github.com/mit-wu-lab/learning-to-delegate`.

**Negative Social Impact.** Enabling more efficient solutions of large-scale VRPs may exhibit negative externalities such as inducing additional traffic from delivery vehicles and centralizing services that pose a stronger competition to brick-and-mortar retail.

# 8 Acknowledgements

This research was supported by MIT Indonesia Seed Fund, US DOT DDETFP, and the MIT-IBM Watson AI Lab. The authors are grateful to the anonymous reviewers for detailed comments that substantially improved the article. The authors acknowledge the MIT SuperCloud and Lincoln Laboratory Supercomputing Center for providing (HPC, database, consultation) resources that have contributed to the research results reported within this paper. We also thank Zongyi Li for helpful discussions and technical advice throughout the project.

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
