# A Appendix

## Contents

## A.1 Experiment Setup

We discuss the full experimental setup for training and evaluating the subproblem selector on the uniform CVRP distribution in this section while reserving any minor modifications for the clustered CVRP distributions, real-world CVRP distribution, VRP variants, and ablation studies for Appendix A.2, A.3, A.4, and A.5, respectively.

**Uniform CVRP distribution.** We generate CVRP instances of size $N = 500, 1000, 2000,$ and 3000 following the same distribution of cities locations and demands as previous works [29, 20, 27, 21]: the depot and cities' $(x, y)$ locations are sampled uniformly from $[0, 1]^2$, each city's $d$ is sampled uniformly from $\{1, 2, ..., 9\}$, and each vehicle has capacity $C = 50$. For each $N$, we sample $n_{\text{train}} = 2000$, $n_{\text{val}} = 40$, and $n_{\text{test}} = 40$ problem instances for the training, validation, and test set.

Table 3: **Uniform CVRP parameters.**

| | |
|---|---|
| Choices of $N$ | $\{500, 1000, 2000, 3000\}$ |
| Dist. of depot location | $\mathcal{U}([0,1]^2)$ |
| Dist. of city location | $\mathcal{U}([0,1]^2)$ |
| Dist. of city demand | $\mathcal{U}(\{1, \ldots 9\})$ |
| Vehicle capacity | 50 |

Table 4: **Data splits.**

| | |
|---|---|
| Training | 2000 instances |
| Validation | 40 instances |
| Test | 40 instances |

**Initialization solution.** Our iterative learning-to-delegate framework (Figure 1) requires a feasible initial solution $X_0$. To generate $X_0$ for each problem instance, we employ a fixed initialization scheme in which the space is partitioned into 10 equally-sized angular sectors radiating outward from the depot. We then run the LKH-3 solver for a brief 100 steps on each partition to produce initial routes. Initialization on average takes 6 seconds for $N = 500$, 18 seconds for $N = 1000$, 50 seconds for $N = 2000$, and 94 seconds for $N = 3000$ on a single CPU. Our initialization scheme is similar to the sweep-based algorithm proposed by Renaud and Boctor [32], which is a commonly used initialization scheme for iterative VRP solvers. Unless stated otherwise, we use the same initialization

Table 5: **Best architecture hyperparameters.**

| | |
|---|---|
| Input dimension | 3 |
| Base model | Transformer |
| Model dimension $d_{\text{model}}$ | 128 |
| Number of heads $n_{\text{heads}}$ | 8 |
| Number of layers $n_{\text{layers}}$ | 6 |
| Feed-forward dimension $d_{\text{ff}}$ | 512 |
| Activation | ReLU |
| Layer Normalization | Yes |
| Dropout | 0 |

Table 6: **Best training hyperparameters.**

| | |
|---|---|
| Optimizer | Adam |
| Learning rate | 0.001 |
| Learning rate schedule | Cosine |
| Batch size | 512 |
| Number of gradient steps | 40000 |
| GPU | NVIDIA V100 |

scheme in all methods to fairly compare each method's ability to improve from a rough initialization. We additionally explore the effect of initialization quality on our method in Appendix A.5.4.

**Training data generation.** We generate the training set by running our iterative framework and selecting subproblems as follows: for each instance $P$ with initial solution $X$ in the training set, we run the subsolver on each $S \in \mathcal{S}_{k,\text{local}}$ to compute the subsolution $X_S$ and improvement $\delta(S)$. We update the solution from $X$ to $X'$ with $\arg\max_S \delta(S)$ then repeat the process on $X'$ for a fixed number of steps $D_{\text{train}} = 30$ of our iterative framework. For the uniform CVRP distribution, for each route $r$ in the current solution $X$, we construct $\mathcal{S}_{k,\text{local}}$ with $k = 10$ nearest routes. We compute the subsolution by running the LKH-3 subsolver for 500 steps, which takes around 6.7 seconds on a single CPU for a $k = 10$ subproblem composed of around 100 cities. We concatenate training data from multiple $N$'s to train the subproblem regression model. As mentioned in the main text, most subproblems remain unchanged from $X$ to $X'$, so we do not repeat unchanged subproblems in our generated training set; therefore, the number of unique subproblems in our training set may be around 3-8 times smaller than the total number of non-unique subproblems $n_{\text{train}} D_{\text{train}} \mathbb{E}[R] \approx n_{\text{train}} D_{\text{train}} \frac{N}{\mathbb{E}[d]}$ in the training set, where $\mathbb{E}[R]$ is the average number of routes in a solution.

As we find that the performance of the subproblem regression model trained on the concatenated $N = 500$ and 1000 data offers satisfactory performance, even when transferred to $N = 2000$ and 3000, we do not collect training data for the latter two cases. Collecting training data is the most computationally intensive component of our work, and the computation time increases with larger $N$, larger $k$, and larger $D_{\text{train}}$. By restricting training data collection to $N = 500$ and 1000, $k = 10$, and $D_{\text{train}} = 30$, the total time taken is around 10 hours on 200 CPUs.

**Input features.** To convert each subproblem $S$ into the input to our Transformer subproblem selector, we create a 3-dim input feature vector for each city consisting of the city's $(x, y)$ locations (centered around the depot's $(x, y)$ location) and the demand $d$ (rescaled by the capacity).

**Architecture and training hyperparameters.** Our best Transformer model uses model dimension $d_{\text{model}} = 128$, $n_{\text{heads}} = 8$ heads, $n_{\text{layers}} = 6$ layers, feed-forward dimension of $d_{\text{ff}} = 512$, ReLU activation, layer normalization, and no dropout. We train with Adam optimizer with a learning rate of 0.001, cosine learning rate decay, and a batch size of 2048 for 40000 gradient steps. All hyperparameters are selected on the validation set and frozen before evaluating on the test set.

**Baseline learning methods.** In Table 1, we compare our methods with two learning baselines: Attention Model (AM) [20] and NeuRewriter [7]. We use the pre-trained AM model provided by the author: https://github.com/wouterkool/attention-learn-to-route. The model is trained on problems of size 100 following the same distribution as ours. The model has two modes of prediction: greedy and sampling. We perform 1280 samples per instance, yet find the sampling performance worse than the greedy result. While the original paper reports better sampling performance than greedy when $N \leq 100$, we believe that AM sampling suffers from the fact that the solution space is exponentially larger with our large problem instance sizes. Therefore, any small stochasticity and imperfections from AM sampling may autoregressively compound much more in our setting than in previous AM settings. We use a single NVIDIA V100 GPU with 20 CPUs for evaluation.

For NeuRewriter, we re-train the network on problems of size 100 using the code provided by the author: `https://github.com/facebookresearch/neural-rewriter`, as no pre-trained model is available. Similarly to AM, we choose size 100 since it is the largest training problem size from the paper. We run into memory and fitting issue when trying to rerun their code on larger problem sizes, so we do not report the numbers here. We use a single NVIDIA V100 GPU with 20 CPUs to perform the evaluation.

## A.2 Clustered and Mixed CVRP

**Clustered CVRP distribution.** We generate clustered CVRP distribution by modifying the distribution of city locations within the uniform CVRP distribution specified in Appendix A.1. Given the number of clusters $n_c$, we first generate $n_c$ cluster centroids by sampling from $\mathcal{U}([0.2, 0.8]^2)$ as the $(x, y)$ locations. Then, we generate the city nodes by first sampling a centroid uniformly at random, and then sampling the $(x, y)$ location normally distributed with the mean at the centroid and standard deviation 0.07. The $(x, y)$ locations are clipped within the $[0, 1]^2$ box.

**Mixed CVRP distribution.** For the mixed distributions, we sample half of the city nodes according to the uniform distribution, and the other half according to a clustered distribution with $n_c$ centers. This is the common practice used in standard CVRP benchmark datasets [44]. Note that we generate our new dataset instead of using the benchmark datasets because the CVRP benchmark datasets include mostly small scale $N \leq 500$ instances and have very few instances with $N \geq 100$.

**Dataset composition.** As we experiment over all combinations of $N \in \{500, 1000, 2000\}$, $n_c \in \{3, 5, 7\}$, and $m \in \{\text{Clustered}, \text{Mixed}\}$, we generate smaller training, validation, and test set sizes with 500, 10, and 10 instances respectively for each combination. We visualize one clustered and one mixed instance in Figure 3.

**Training and evaluation.** Like in uniform CVRP, we do not collect the training set for the $N = 2000$ instances and concatenate the training set for all remaining combinations to create a single training set of $2 * 3 * 2 * 500 = 6000$ problem instances. We use identical architecture and training hyperparameters as listed in Appendix A.1, as we find that these hyperparameters perform reasonably well. We report results on the test set of 10 instances for each of the $3 * 3 * 2 = 18$ settings separately.

**Additional results.** We include the full evaluation results on the test set for all clustered and mixed distributions. Figure 18 contains results for $N = 500$, Figure 19 contains results for $N = 1000$, and Figure 20 contains results for $N = 2000$. As mentioned in the above paragraph, a single trained model for our method is evaluated on all combinations of $(N, n_c, m)$, which demonstrates superior speedup and improvement (when run for a reasonable time period) over the Random baseline. To save space, we also include results with the HGS subsolver in the same set of plots.

## A.3 Out-of-distribution CVRPs

Here we describe the CVRP distributions and full results supporting the out-of-distribution generalization performance of our trained uniform CVRP and clustered CVRP models, with full training specifications in Appendix A.1 and Appendix A.2 respectively.

### A.3.1 Uniform CVRP with $N = 3000$

We do not describe the setup here as it was previously described with the uniform CVRP distribution in Appendix A.1. In Figure 9, we see that the model trained on the clustered CVRP demonstrates slightly better improvement than the model trained on the uniform CVRP; otherwise the model performances are similar.

### A.3.2 Real-world CVRP

Excitingly, we see robust generalization performance of our previously trained models to problem instances from an unseen real-world CVRP distribution derived from the large-scale real-world CVRP dataset generated by Arnold et al. [4], which is open source at `https://antor.uantwerpen.be/xxlrouting/`.

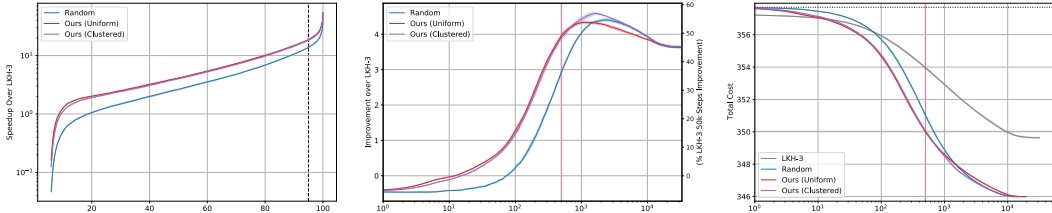

Figure 9: **Uniform CVRP distributions,** $N = 3000$. Graphs of speedup (left, higher is better), improvement (middle, higher is better), and total cost (right, lower is better). The left graph is the same as the left graph in Figure 6. The right vertical axis in the improvement graph (middle) indicates the improvement over LKH-3 as a percentage of the total improvement that LKH-3 attains over 30k steps. The horizontal dotted black line in the total cost graph (right) indicates the initial solution quality for all methods. The vertical dashed black line in the speedup graph (left) and the horizontal LKH-3-colored line in the total cost graph (right) indicate the 95% LKH-3 solution quality. The overlapping colored vertical lines in the improvement graph (middle) and the total cost graph (right) indicate computation times required for the corresponding learning-based methods to reach the aforementioned solution quality.

Table 7: **Instance sizes** $N$ **of original real-world instances from Arnold et al. [4].** Instances are subsampled without replacement to $N = 2000$ before feeding into our model.

| Region | Centralized Depot | Eccentric Depot |
|---|---|---|
| Leuven | 3000 | 4000 |
| Antwerp | 6000 | 7000 |
| Ghent | 10000 | 11000 |
| Brussels | 15000 | 16000 |
| Flanders | 20000 | 30000 |

**Original real-world CVRP distribution.** The original dataset contains 10 instances representing 5 different Belgium regions: Leuven, Antwerp, Ghent, Brussels, and Flanders. For each region, two instances are provided: one with a depot located at the center of all cities and the other with an eccentric depot located in the outskirts of the regions. We list the original sizes of each instance in Table 7 and visualize them in Figure 10.

This dataset focuses on long-haul deliveries, with the average solution route length ranging from 14.8 to 117.2. The authors randomly sample demand for each node from $\{1, 2, 3\}$ with probabilities 0.5,

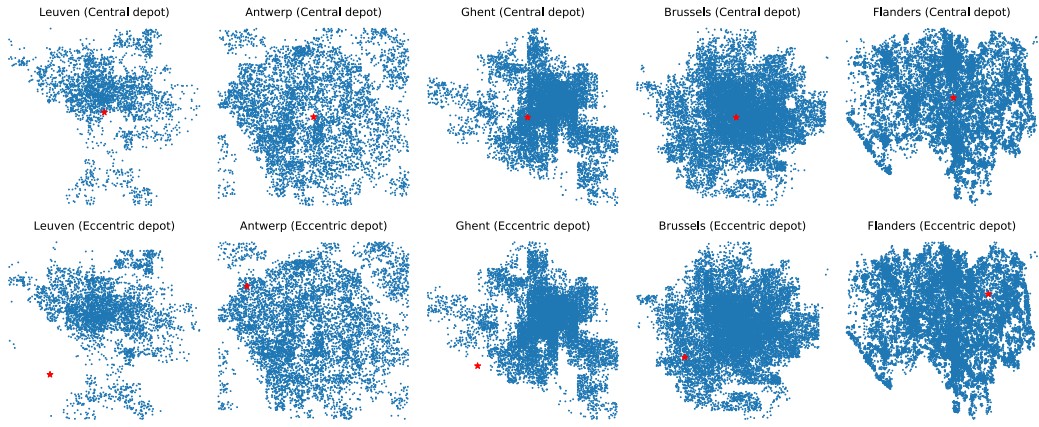

Figure 10: **Original real-world CVRP instances.** Five instances (top row) have centralized depots and five (bottom row) have eccentric depots.

0.3, and 0.2 respectively, and the capacity of the vehicle ranges from 20 to 50 for central depot and ranges from 100 to 200 for eccentric depot.

**Our real-world CVRP dataset.** To apply our pre-trained models, we first convert the original real-world CVRP to be reasonably close to our training distributions. To generate each instance in our real-world CVRP distribution from an instance in the original dataset, we subsample the original instance to $N = 2000$ without replacement, resample each demand from our demand distribution $\mathcal{U}(\{1, 2, \ldots, 9\})$, then set the vehicle capacity to $C = 50$. For each original instance, we generate 5 instances from our real-world CVRP distribution. Therefore, our validation and test sets contain 50 instances in total. One instance from our test set is visualized in Figure 3.

**Additional results.** We show the full transfer performance of the models trained on uniform and clustered CVRP to our real-world CVRP distribution in Figure 11. We see that the model trained on the clustered CVRP distributions significantly outperforms both the model trained on the uniform CVRP distribution and the Random baseline, whereas the model trained on the uniform CVRP distribution sometimes performs worse than the Random baseline. Encouragingly, the clustered model shows strong generalization performance in CVRP distributions which may be of practical value in the real world.

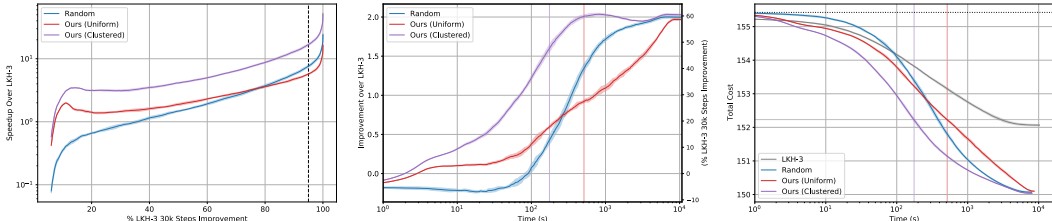

Figure 11: **Our real-world CVRP distribution.** Graphs of speedup (left, higher is better), improvement (middle, higher is better), and total cost (right, lower is better). The right vertical axis in the improvement graph (middle) indicates the improvement over LKH-3 as a percentage of the total improvement that LKH-3 attains over 30k steps. The horizontal dotted black line in the total cost graph (right) indicates the initial solution quality for all methods. The vertical dashed black line in the speedup graph (left) and the horizontal LKH-3-colored line in the total cost graph (right) indicate the 95% LKH-3 solution quality. The colored vertical lines in the improvement graph (middle) and the total cost graph (right) indicate computation times required for the corresponding learning-based methods to reach the aforementioned solution quality.

## A.4 VRP Variants

Here we provide the full definition, experimental setup, and results for VRP variants discussed in Section 6.5: CVRPTW and VRPMPD.

### A.4.1 CVRPTW (Capacitated Vehicle Routing Problem with Time Windows)

**Problem formulation.** We use the standard CVRPTW formulation found in Gehring and Homberger [11] and Solomon [37]. In additional to all CVRP constraints, each city node $i$ has a time window $[e_i, l_i]$, the earliest and latest time to visit city $i$, respectively, and a service time $s_i$. The depot has a time window $[e_0, l_0]$ but no service time. Each vehicle departs the depot at time $e_0$ and follows a route $r$ to satisfy the additional time window constraint. Let $i$ and $j$ be consecutive cities on route $r$, and let $b_i^r$ denote the arrival time at city $i$ of the vehicle serving route $r$. Then the departure time at city $i$ is $b_i^r + s_i$ and the arrival time at the next city $j$ is $b_j^r = \max\{e_j, b_i^r + s_i + t_{ij}\}$ where $t_{ij}$ is the travel time that equals the Euclidean distance from $i$ to $j$. This specifies that if a vehicle arrives too early at $j$, then it needs to wait until the start time $e_j$ of $j$. Meanwhile, the route becomes infeasible if the vehicle arrives at $j$ later than $l_j$. Finally, the depot's time window requires the route to travel back to the depot before $l_0$. The number of vehicles is unconstrained. The objective is the same as in CVRP, i.e. minimizing the total Euclidean edge costs of all the routes, which is the standard objective used in the literature [19, 17]. The time window constraint imposed by this variant is related to other combinatorial optimization problems such as job shop scheduling [5].

**Problem distribution.** We generate CVRPTW instances following a similar procedure as in Solomon [37]: we first generate CVRP instances following the same uniform location and demand distribution as described in Section A.1. For the time window constraint, we set the time window for the depot as $[e_0, l_0] = [0, 3]$, and the service time at each $i$ to be $s_i = 0.2$. We further set the time window for city node $i$ by (1) sampling the time window center $c_i \sim \mathcal{U}([e_0 + t_{0,i}, l_0 - t_{i,0} - s_i])$, where $t_{0,i} = t_{i,0}$ is the travel time, equaling the Euclidean distance, from the depot to node $i$; (2) sampling the time window half-width $h_i$ uniformly at random from $[s_i/2, l_0/3] = [0.1, 1]$; (3) setting the time window for $i$ as $[\max(e_0, c_i - h_i), \min(l_0, c_i + h_i)]$.

**Training data generation.** Unlike for CVRP distributions, we generate training data using $k = 5$ routes per subproblem instead of $k = 10$; doing so does not change the number of subproblems at each step, but significantly speeds up the generation process for two reasons: 1) running the subsolver on each subproblem takes around 3.4 seconds instead of 6.7 seconds for $k = 10$ and 2) updating current $X$ with the subsolution $X_S$ for a smaller subproblem $S$ means that there will be more repeated subproblems from $X$ to $X'$ as other subproblems are less likely to be affected. Therefore, in around 10 hours total on 200 CPUs, we are able to generate training data with 2000 instances per $N$ and $D_{\text{train}} = 40, 80$, and 160 for $N = 500, 1000$, and 2000, respectively. An ablation study on the effect of $k$ in uniform CVRP is provided in Appendix A.5.2.

**Additional results.** We show the full CVRPTW results in Figure 14, which also includes a comparison between subproblem regression and subproblem classification. Interestingly, while our method offers significant speedup over baseline methods and offers significant improvement over LKH-3 (relative to LKH-3's total improvement over 30k steps) when run for a reasonable amount of time, the improvement over the LKH-3 baseline diminishes when run for a very long time. Indeed, our method terminates at a slightly worse total cost than LKH-3 for $N = 500$ when run for a very long time. However, as we show in the ablation over subproblem size $k$ in Appendix A.5.2, we see that using $k = 10$ tends to offer better eventual improvement in uniform CVRP, so the diminishing improvement effect may be partially attributed to the fact that we use $k = 5$ for the CVRPTW experiments.

### A.4.2 VRPMPD (Vehicle Routing Problem with Mixed Pickup and Delivery)

**Problem formulation.** This VRP variant models the backhauling problem where the vehicles are both delivering items to cities from the depot and picking up items from cities to bring back to the depot. We use the standard VRPMPD formulation as in Salhi and Nagy [33], where each city $i$ either has a pickup order with load $p_i$ or delivery order with load $d_i$. Each vehicle with a capacity $C$ starts by loading all delivery orders along the route. Each time the vehicle visits a delivery city $i$, the vehicle's load decreases by $d_i$; conversely, each time the vehicle visits a pickup city $j$, the vehicle's load increases by $p_j$. A valid route requires that the vehicle's load is no greater than $C$ at every point along the route. Therefore, this variant imposes a more complicated capacity constraint than CVRP as the order of city visitation affects the feasibility of a route.

**Problem distribution.** We generate VRPMPD instances following a similar procedure as in Salhi and Nagy [33]: we take the CVRP instances following the same uniform location and demand distribution as defined in Appendix A.1, and we randomly assign half the cities pickup orders with $p_i \sim \mathcal{U}(\{1, 2, \ldots, 9\})$ and the other half delivery orders with $d_i \sim \mathcal{U}(\{1, 2, \ldots, 9\})$. We set the vehicle's capacity to be $C = 25$ instead of 50 (as done for other VRPs) because each route in VRPMPD visits roughly 2x more cities than those in CVRP or CVRPTW with the same capacity, as the vehicle gains empty space after visiting the cities with delivery orders and can serve cities with pickup orders afterward.

**Training data generation.** We use the same $k = 5$ data generation process as described in Appendix A.4.1, but for VRPMPD instead of CVRPTW.

**Additional results.** We show the full VRPMPD results in Figure 15, which also includes a comparison between subproblem regression and subproblem classification. Similar to the behavior in CVRPTW, while our method offers significant speedup over baseline methods and offers significant improvement over LKH-3 (relative to LKH-3's total improvement over 30k steps) when run for a reasonable amount of time, the improvement over the LKH-3 baseline diminishes when run for a very long time. However, as we show in the ablation over subproblem size $k$ in Appendix A.5.2, we see

that using $k = 10$ tends to offer better eventual improvement in uniform CVRP, so the diminishing improvement effect may be partially attributed to the fact that we use $k = 5$ for the VRPMPD experiments.

### A.5  Ablation Studies

#### A.5.1  Regression vs Classification for Subproblem Selection

While we focus on regression in Section 5, here we discuss how to frame subproblem selection as classification, and compare results and discuss trade-offs with regression-based subproblem selection.

**Subproblem selection as classification.**  The goal of a classification-based subproblem selector is to select the subproblem which results in the best immediate improvement when solved by the subsolver.

**Label transformation.**  Optimally, we would like our subproblem selector to select the subproblem with the best immediate improvement. However, since we want to avoid overpenalizing the subproblem selector for putting probability mass on a slightly suboptimal selection, we construct a softmax soft label based on $\delta(S)$ with temperature parameter $\tau$ rather than a one-hot hard label, which corresponds to $\tau \to 0$

$$\ell(S) = \frac{e^{\delta(S)/\tau}}{\sum_{S' \in \mathcal{S}_{k,\text{local}}} e^{\delta(S')/\tau}} \tag{3}$$

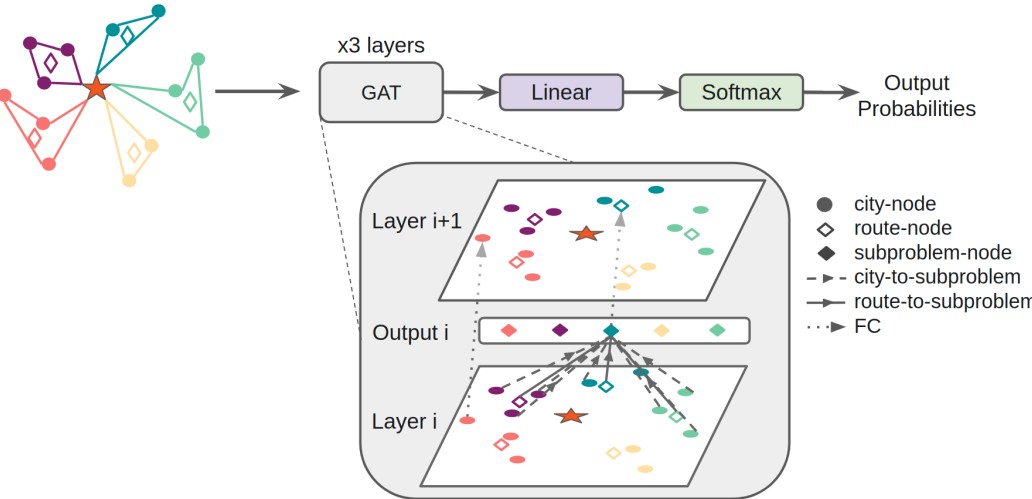

Figure 12: **Our Graph Attention Network (GAT) architecture**.  At each step of our iterative framework (Figure 1(c)), we featurize the problem instance $P$ with current solution $X$ into city-node (filled circle) and route-node (empty diamond) features and city-to-subproblem and route-to-subproblem edges. The city-node input features at the $i$th graph attention layer is passed through a FC network to obtain the input city-node features at the $i + 1$th graph attention layer. The subproblem-node (filled diamond) output features of the $i$th graph attention layer is passed through a FC network to obtain the input route-node features at the $i + 1$th graph attention layer. The final graph attention layer's subproblem-node output features are fed into a linear layers with softmax activation to obtain the classification probabilities $p_\phi(S)$ for each subproblem.

**Network architecture.**  An important consequence of subproblem classification is that all subproblems must be considered jointly. This requires a different architecture than presented in Section 5. Intuitively, in order to consider which subproblem is *best*, the other subproblems must be taken into account. In the regression context, a local measure of improvement is all that is needed. More precisely, as subproblem classification inherently requires feed-forward and back-propagation computation over all subproblems $S \in \mathcal{S}_{k,\text{local}}$ for a single problem $P$, our choice of architecture must

address this constraint to feasibly train with a large enough batch of problems. Unlike in subproblem regression, we cannot simply sample batches of subproblems from the set of all problem instances to fit individually; each computation in classification entails a softmax over $|S_{k,\text{local}}|$ subproblems from the same problem instance. Therefore, our classification-based subproblem selector consists of a Graph Attention Network (GAT) backbone [46], which shares computation and memory between overlapping subproblems in the same problem instance, permitting the use of large batches of problem instances $P$ to increase training stability.

We illustrate the nodes and edges of the graph attention in Figure 12. Given a problem instance $P$, we define three types of graph nodes: one *city-node* per city in $P$, one *route-node* per route $r$ in the current solution $X$, and one subproblem node per $S \in \mathcal{S}_{k,\text{local}}$. We define two types of directed edge connections: a *city-to-subproblem* edge connects a city-node to a subproblem-node if the city is in the subproblem and a *route-to-subproblem* edge connects a route-node to a subproblem-node if the route is in the subproblem. The input city-node features is fed into a fully connected (FC) network to obtain the input city-node features of the next graph attention layer, while the output subproblem-node features is fed into another FC network to obtain the input route-node features for the next graph attention layer, as each subproblem $S$ corresponds to a single route $r$ by construction as described in Subsection 4.1. The subproblem-node features of the final graph attention layer are fed into a linear layer with softmax activation to obtain classification probabilities.

**KL divergence loss.** We minimize the KL divergence between the ground-truth soft labels $\ell(S)$ and the subproblem selector output probability distribution $p_\phi(S)$, where $\phi$ is the parameters of the subproblem selector

$$L(\phi; P) = D_{\text{KL}}(\ell \| p_\phi) = \sum_{S \in \mathcal{S}_{k,\text{local}}} \ell(S) \log \left( \frac{\ell(S)}{p_\phi(S)} \right). \tag{4}$$

One possible advantage of the KL divergence loss is that it places the most weight on subproblems with high $\ell(S)$, which may help the neural network focus on a few subproblems with high immediate improvements rather than the many subproblems with low immediate improvement. In contrast, the Huber loss used in subproblem regression places equal emphasis on fitting subproblems with low and high immediate improvements.

**Classification setup.** While we previously describe the setup of the regression model in Appendix A.1, here we describe the additional modifications made in the classification setup compared with the regression setup.

1. While we do not need to collect separate training data from the regression setup, we need to process the label to be $\ell(S)$ rather than $\delta(S)$.

2. In addition to extracting 3-dim feature vector for each city in $P$, we also extract 8-dim feature vector for each route in the current solution $X$ consisting of: the average, max, and min $(x, y)$ locations of cities in the route (shifted by the depot's $(x, y)$ location); the sum of demands of cities in the route; and the route's distance.

3. We find the best GAT architecture hyperparameters to be a model (hidden) dimension of 128, 1 attention head, and 3 graph attention layers.

4. As the classification network does not handle multiple problem sizes (e.g. $N = 500$ and $N = 2000$) naturally due to large differences in the number of subproblems, we train each classification model on data from a single problem instance size. However, this is not a fundamental limitation, and future works may explore training a subproblem classification network on multiple problem sizes.

5. We train our best classification models with batch sizes of $256, 256$, and $512$ *problem instances* respectively for $N = 500, 1000, 2000$. We emphasize that the batches here are batches of *problem instances*, where each problem instance graph inherently contains $|\mathcal{S}_{k,\text{local}}|$ subproblems as illustrated earlier in Figure 12, rather than batches of *subproblems* as in the regression setup.

6. Training takes around 6 hours for $N = 500$, 12 hours for $N = 1000$, and 24 hours for $N = 2000$ on a single NVIDIA V100 GPU.

Table 8: **Training time of subproblem regression and subproblem classification.** Subproblem regression models are simultaneously trained over all problem sizes where subproblem classification models are trained on individual problem sizes. Training times are similar for subproblems with size $k = 5$ and $k = 10$, so we do not distinguish between the two cases in this table. Note that we do not use $N = 2000$ data for subproblems with size $k = 10$ for training, as explained in Appendix A.1, but we do use $N = 2000$ data for subproblems with size $k = 5$ for training, as explained in Appendix A.5.2.

| $N$ | **Regression** | **Classification** |
|---|---|---|
| 500 | | 6 hours |
| 1000 | **6 hours total** | 12 hours |
| 2000 | | 24 hours |

Table 8 summarizes a key advantage of subproblem regression over subproblem classification: subproblem regression is much faster to train and only requires a single trained model over all problem instance sizes.

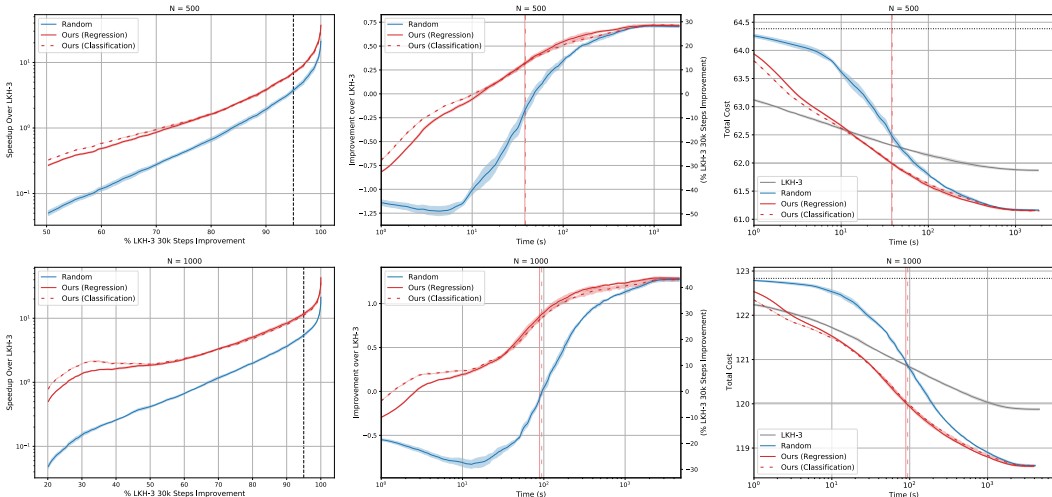

Figure 13: **Uniform CVRP distribution, regression vs classification.** Graphs of speedup (left column, higher is better), improvement (middle column, higher is better), and total cost (right column, lower is better). The right vertical axis in each improvement graph (middle column) indicates the improvement over LKH-3 as a percentage of the total improvement that LKH-3 attains over 30k steps. We use loosely dashed lines to indicate classification results. The horizontal dotted black line in the total cost graph (right) indicates the initial solution quality for all methods. The vertical dashed black line in the speedup graph (left) and the horizontal LKH-3-colored line in the total cost graph (right) indicate the 95% LKH-3 solution quality. The styled vertical lines in the improvement graph (middle) and the total cost graph (right) indicate computation times required for the corresponding learning-based methods to reach the aforementioned solution quality.

**Comparison on uniform CVRP.** We compare the performance of subproblem regression and subproblem classification for $N = 500$ and $1000$ uniform CVRP and $k = 10$. Despite requiring significantly less training time, we observe that subproblem regression does not suffer a loss in performance against the more specialized subproblem classification, as seen in Figure 13. Thus, we chose to focus on subproblem regression in this paper as it offers generalization over multiple problem sizes while enjoying significantly less training time.

**Comparison on VRP variants.** We compare the performance of subproblem regression and subproblem classification for CVRPTW and VRPMPD instances of size $N = 500, 1000$ and $2000$ with $k = 5$. Interestingly, we see that subproblem classification somewhat outperforms subproblem regres-

sion in Figure 14 and Figure 15. Therefore, future works may benefit from trying both regression and classification and possibly adapting subproblem classification to train over multiple problem sizes.

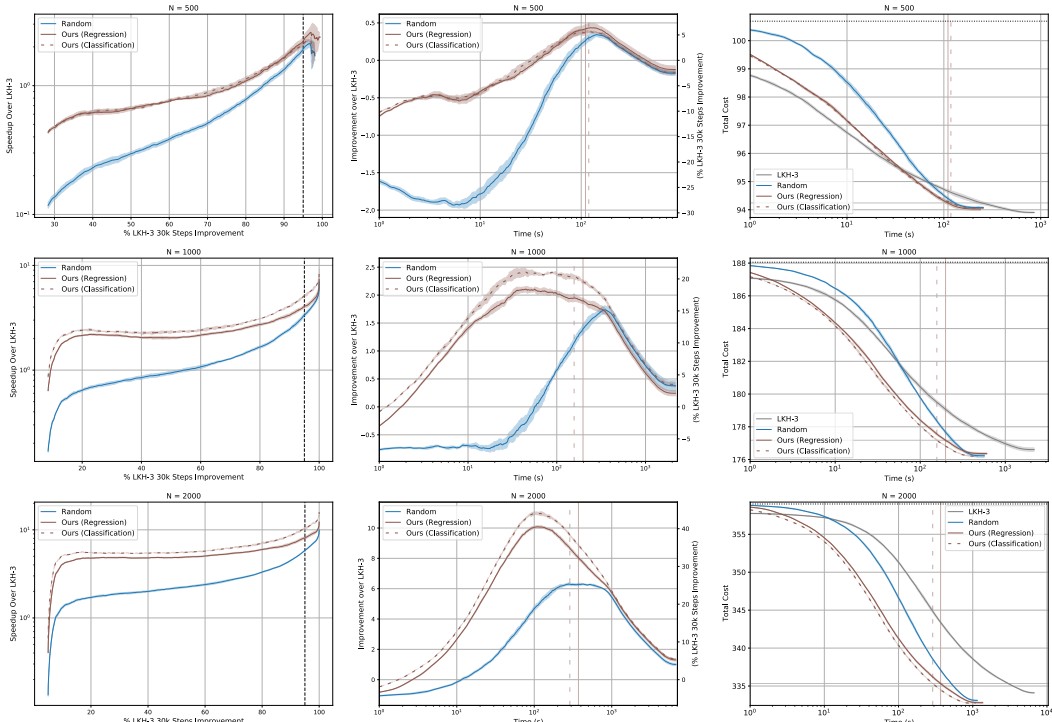

Figure 14: **Uniform CVRPTW distributions.** Graphs of speedup (left column, higher is better), improvement (middle column, higher is better), and total cost (right column, lower is better). The right vertical axis in each improvement graph (middle column) indicates the improvement over LKH-3 as a percentage of the total improvement that LKH-3 attains over 30k steps. We use loosely dashed lines to indicate classification results. The horizontal dotted black line in the total cost graph (right) indicates the initial solution quality for all methods. The vertical dashed black line in the speedup graph (left) and the horizontal LKH-3-colored line in the total cost graph (right) indicate the 95% LKH-3 solution quality. The styled vertical lines in the improvement graph (middle) and the total cost graph (right) indicate computation times required for the corresponding learning-based methods to reach the aforementioned solution quality.

### A.5.2 Subproblem Size $k$ and Asymptotic Behavior in Uniform CVRP

We explore the effect of the subproblem size $k$, which specifies the number of nearest neighbor routes used to create each subproblem, and show that the choice of $k$ may affect the asymptotic behavior of our iterative framework. We compare results between $k = 5$ and $k = 10$, using the same $k = 5$ data generation process as described in Appendix A.4.1 for uniform CVRP (not CVRPTW).

**Experimental results.** As shown in Figure 16, our method and the Random baseline with $k = 5$ may offer additional speedup over their $k = 10$ counterparts. However, they tend to converge earlier to worse eventual solution qualities when run for a very long time, though still better than the eventual solution qualities of LKH-3. These results suggest that subproblem selection incorporating both $k = 5$ and $k = 10$ could combine the superior speedup of $k = 5$ with the better eventual solution qualities of $k = 10$.

### A.5.3 CVRP Subsolvers: LKH-3 vs HGS

We seek to demonstrate the independence of our framework from the LKH-3 subsolver by applying our framework with HGS [49, 48] as the subsolver to the problem instances from Appendix A.1 and Appendix A.2. Here we discuss our HGS-related experimental setup and results.

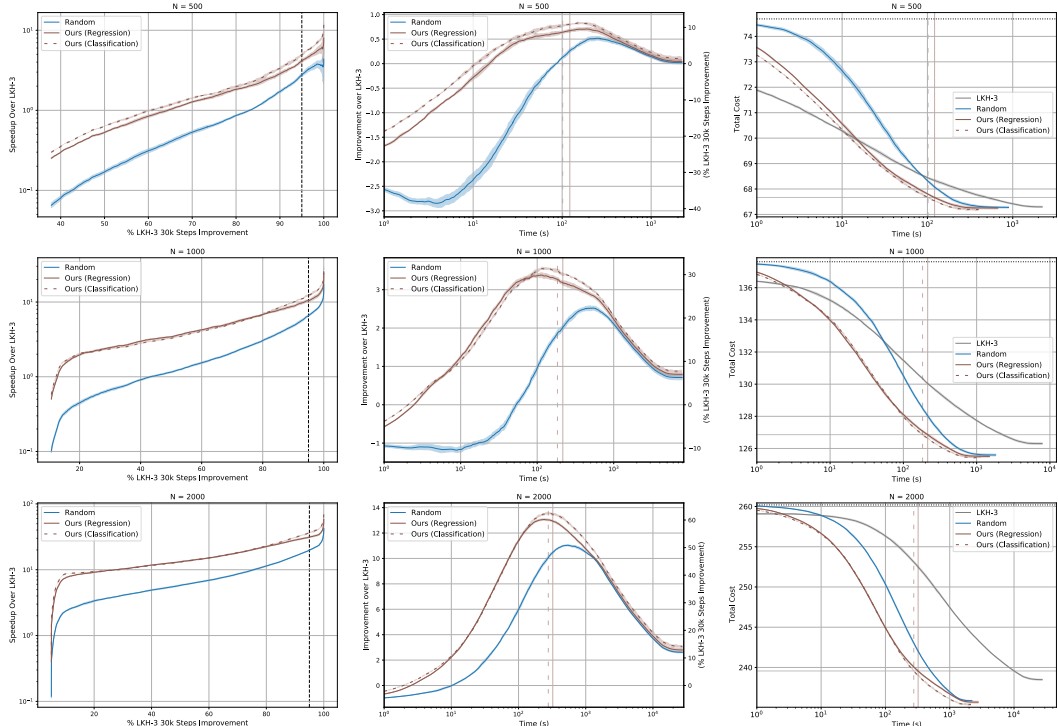

Figure 15: **Uniform VRPMPD distributions.** Graphs of speedup (left column, higher is better), improvement (middle column, higher is better), and total cost (right column, lower is better). The right vertical axis in each improvement graph (middle column) indicates the improvement over LKH-3 as a percentage of the total improvement that LKH-3 attains over 30k steps. We use loosely dashed lines to indicate classification results. The horizontal dotted black line in the total cost graph (right) indicates the initial solution quality for all methods. The vertical dashed black line in the speedup graph (left) and the horizontal LKH-3-colored line in the total cost graph (right) indicate the 95% LKH-3 solution quality. The styled vertical lines in the improvement graph (middle) and the total cost graph (right) indicate computation times required for the corresponding learning-based methods to reach the aforementioned solution quality.

**Initialization solution.** To facilitate comparison with previous results, we use the exact same *LKH-3-based* initialization solutions for all instances as described in Appendix A.1. One note is that the HGS solver does not take an initial solution, so we cannot start running the HGS baseline from the same LKH-3-based initialization as the other methods. To compensate for this limitation, we instead subtract the mean LKH-3-based initialization time from the HGS baseline's total runtime. After this runtime adjustment, we calculate the total HGS improvement by defining the initial HGS solution quality to be the solution quality of our initialization for all other methods. Overall, the initialization time is negligible compared to the total computation time.

**Training data generation.** As the HGS subsolver is significantly faster than LKH-3, we run the subsolver for 1 second on each subproblem (compared to 6.7 seconds for LKH-3 on size $k = 10$ subproblems). This allows us to collect data for $N = 500, 1000,$ and $2000$ and $D_{train} = 80, 160,$ and $320$, respectively, within 10 hours total on 200 CPUs. The remaining uniform and clustered/mixed setup details are described in Appendix A.1 and Appendix A.2, respectively. We collect HGS-based training data for both uniform CVRP and clustered/mixed CVRP, then train a subproblem selector for each data distribution.

**Baseline.** We compute the speedup and improvement of HGS-based subproblem selectors relative to the HGS solver itself. For uniform CVRP, we run the HGS baseline for the same amount of computation time as our LKH-3 baseline: 1800 seconds, 4620 seconds, 8940 seconds, and 30000 seconds for $N = 500, 1000, 2000,$ and $3000$, respectively. However, since there are multiple

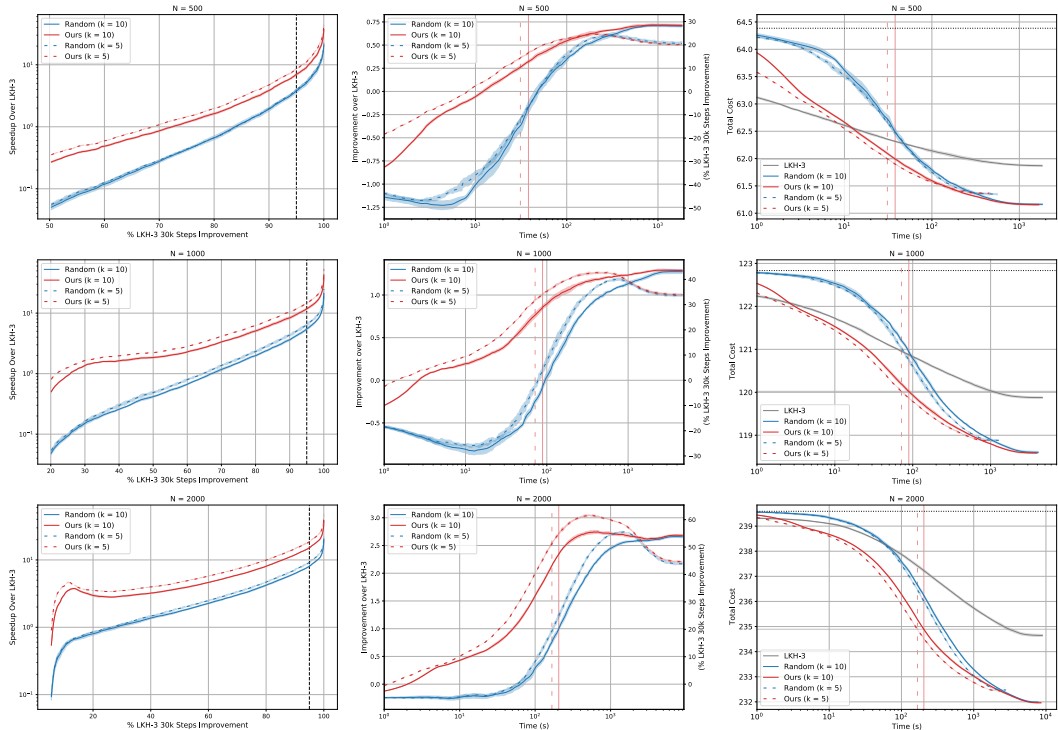

Figure 16: **Uniform CVRP distributions,** $k = 10$ **and** $k = 5$**.** Graphs of speedup (left column, higher is better), improvement (middle column, higher is better), and total cost (right column, lower is better). The right vertical axis in each improvement graph (middle column) indicates the improvement over LKH-3 as a percentage of the total improvement that LKH-3 attains over 30k steps. We use loosely dashed lines to indicate $k = 5$ results. The horizontal dotted black line in the total cost graph (right) indicates the initial solution quality for all methods. The vertical dashed black line in the speedup graph (left) and the horizontal LKH-3-colored line in the total cost graph (right) indicate the 95% LKH-3 solution quality. The styled vertical lines in the improvement graph (middle) and the total cost graph (right) indicate computation times required for the corresponding learning-based methods to reach the aforementioned solution quality.

combinations of $(N, n_c, m)$ for clustered and mixed distributions with different computation times, we run the HGS baseline for 2000 seconds, 5000 seconds, and 10000 seconds for $N = 500, 1000$, and 2000 for every $(N, n_c, m)$.

**Experimental results.** Figure 17 compares the speedup over the subsolver when using either LKH-3 or HGS as the subsolver for uniform CVRP of size $N = 500, 1000, 2000$, and 3000. Similarly, Figures 18, 19, and 20 contain results for mixed and clustered CVRP distributions of size $N = 500, 1000$, and 2000, respectively. We observe that for $N = 500$, our method with the HGS subsolver offers a small speedup over the HGS solver for attaining intermediate range of solution qualities; however, the HGS solver converged to better final solution quality in all cases. This observation is unsurprising, as HGS is designed and calibrated to obtain state-of-the-art solutions for instances of size 500 to 1000. Nevertheless, as the problem size $N$ increases, the speedup of our method over HGS increases, demonstrating the effectiveness of subproblem selection in large-scale problems. For $N = 2000$ and 3000 we are not able to run HGS until full convergence, which would require another order of magnitude more time. Our learned method outperformed the Random baseline in all cases, demonstrating that learning further accelerates subproblem selection.

### A.5.4 Effect of Initial Solution Quality

As our iterative framework relies on an initial solution generated according to Appendix A.1, we explore the effect of poorer, out-of-distribution initial solutions on our subproblem selector for uniform CVRP instances.

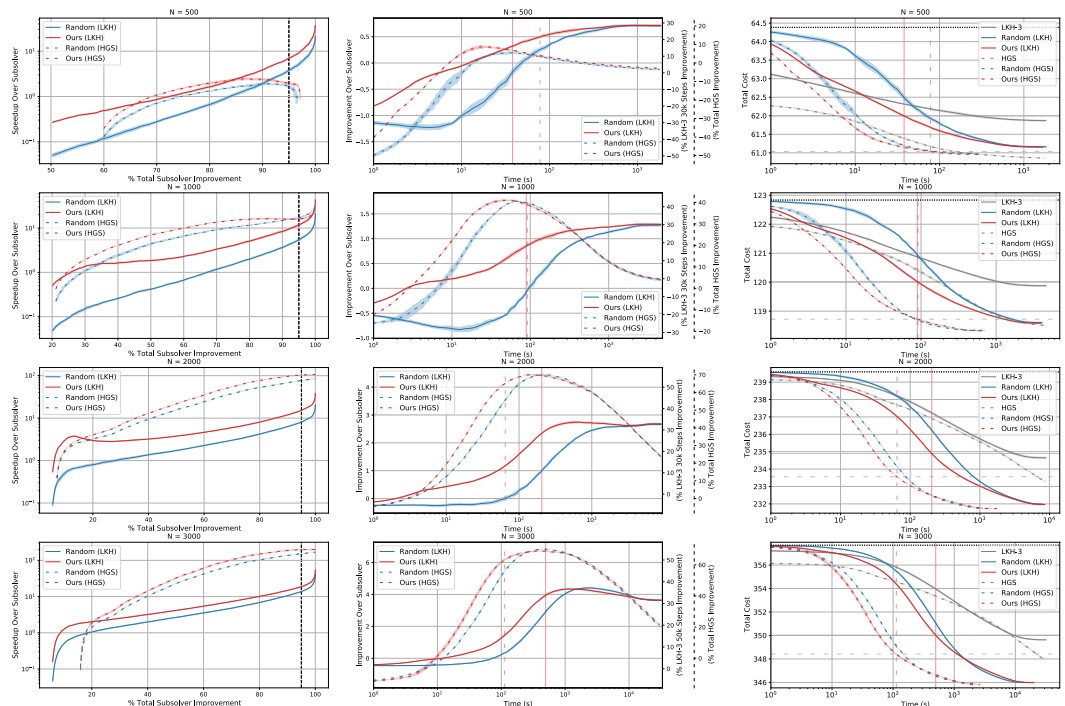

Figure 17: **Uniform CVRP distributions, LKH-3 and HGS subsolvers.** Graphs of speedup (left column, higher is better), improvement (middle column, higher is better), and total cost (right column, lower is better). We use loosely dashed lines to indicate HGS results. The right vertical axes in each improvement graph (middle column) indicates the improvement over the corresponding subsolver as a percentage of the best total improvement that the subsolver baseline attains. The horizontal dotted black line in the total cost graph (right) indicates the initial solution quality for all methods. The vertical dashed black line in the speedup graph (left) and the horizontal subsolver-colored lines in the total cost graph (right) indicate 95% solution qualities for the corresponding subsolvers. The styled vertical lines in the improvement graph (middle) and the total cost graph (right) indicate computation times required for the corresponding learning-based methods to reach the corresponding aforementioned solution qualities.

**Initialization solution.** Let $L$ denote the number of LKH steps run on each partition at initialization. Previously, $L = 100$ as stated in Appendix A.1. To generate initial solutions of poorer quality, we additionally generate initial solutions with $L = 1, 5, 10,$ and $50$. As a last initialization method, which does not use LKH-3 and is denoted $L = 0$, we uniformly randomly chain cities into routes with no consideration for their location. We do not train or finetune any models for this ablation, instead evaluating the performance of our previous uniform CVRP model (trained on $L = 100$ initializations) on all other initialization methods. Note that we calculate the speedup and improvement of all methods with respect to LKH-3 with $L = 100$ initialization.

**Experimental results.** We provide experimental results in Figure 21. We find that for each subproblem selection method (Ours and Random), worse initialization corresponds to somewhat worse speedup and worse improvement over LKH-3 with $L = 100$. Nevertheless, Ours outperforms Random for every $L$ considered, demonstrating that our subproblem selector trained on $L = 100$ data generalizes well to out-of-distribution initial solutions. We are also able to show that subproblem selection with even the worst initialization offers a speedup above 1x for attaining 95% of $L = 100$ LKH-3 solution quality.

### A.5.5 Transformer vs Simpler Architectures

We seek to illustrate the importance of our Transformer regression architecture illustrated in Figure 2 by comparing with subproblem selectors with simpler architectures trained on uniform CVRP.

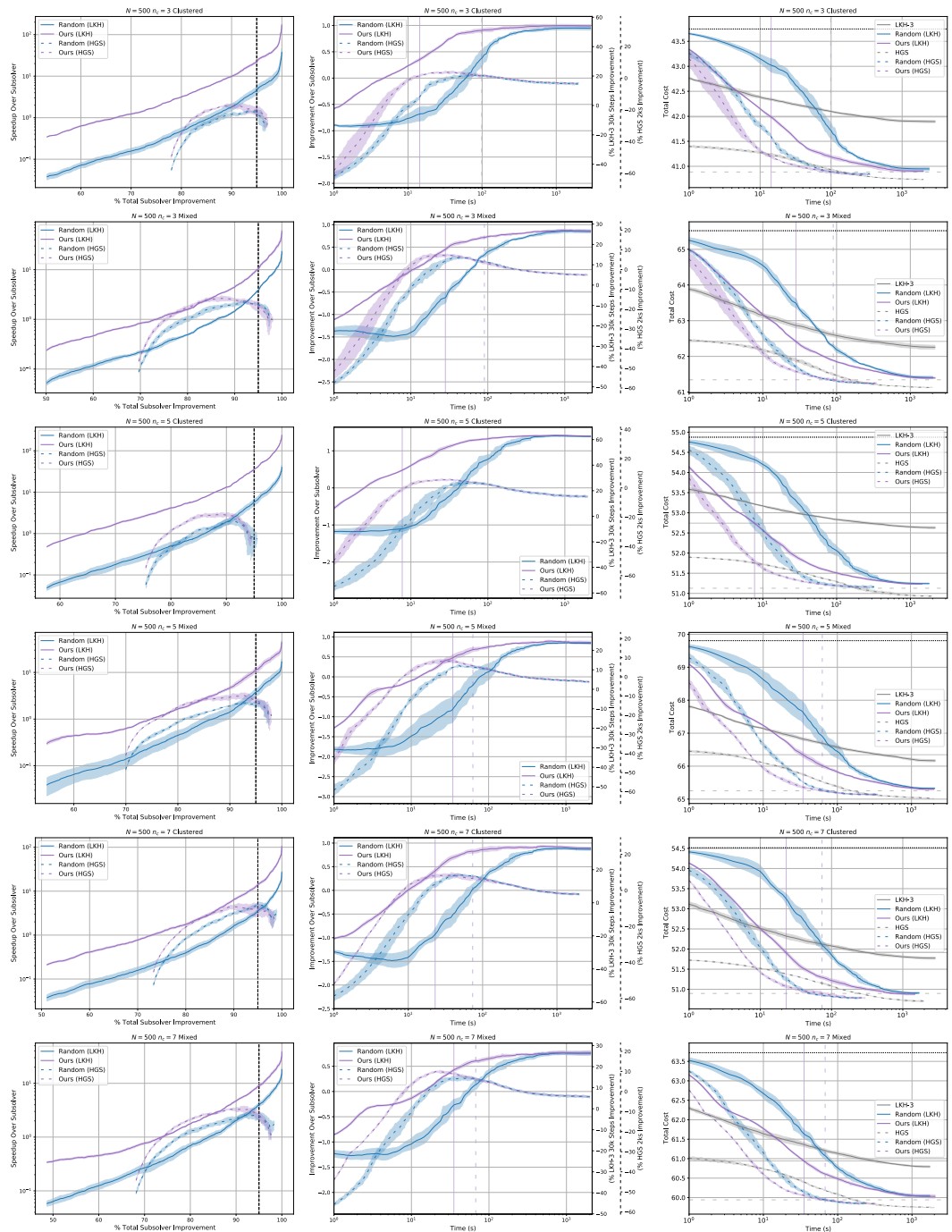

Figure 18: **Clustered and mixed CVRP distributions,** $N = 500$**, LKH-3 and HGS subsolvers.** Graphs of speedup (left column, higher is better), improvement (middle column, higher is better), and total cost (right column, lower is better). We use loosely dashed lines to indicate HGS results. The right vertical axes in each improvement graph (middle column) indicates the improvement over the corresponding subsolver as a percentage of the best total improvement that the subsolver baseline attains. The horizontal dotted black line in the total cost graph (right) indicates the initial solution quality for all methods. The vertical dashed black line in the speedup graph (left) and the horizontal subsolver-colored lines in the total cost graph (right) indicate 95% solution qualities for the corresponding subsolvers. The styled vertical lines in the improvement graph (middle) and the total cost graph (right) indicate computation times required for the corresponding learning-based methods to reach the corresponding aforementioned solution qualities.

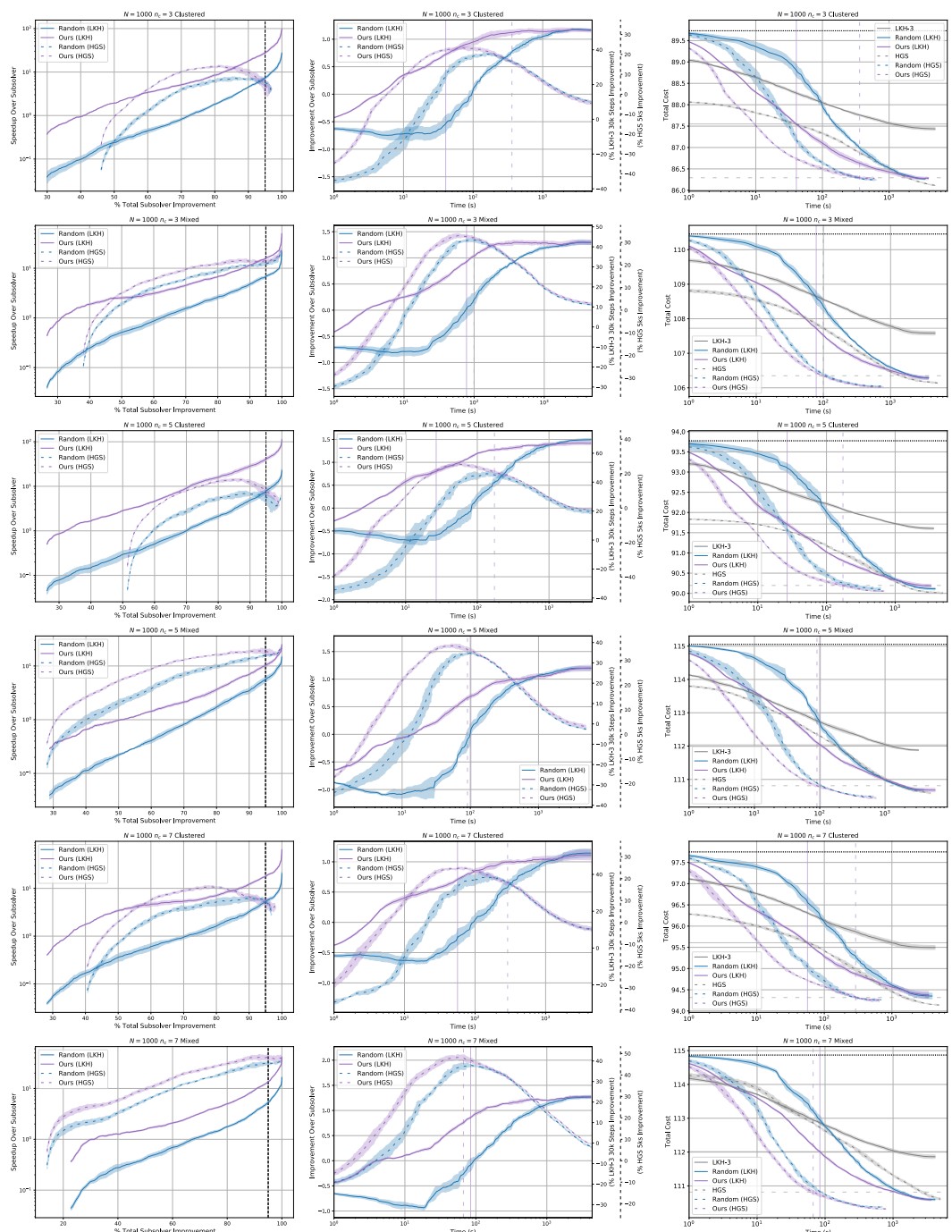

Figure 19: **Clustered and mixed CVRP distributions,** $N = 1000$**, LKH-3 and HGS subsolvers.**
Graphs of speedup (left column, higher is better), improvement (middle column, higher is better),
and total cost (right column, lower is better). We use loosely dashed lines to indicate HGS results.
The right vertical axes in each improvement graph (middle column) indicates the improvement
over the corresponding subsolver as a percentage of the best total improvement that the subsolver
baseline attains. The horizontal dotted black line in the total cost graph (right) indicates the initial
solution quality for all methods. The vertical dashed black line in the speedup graph (left) and the
horizontal subsolver-colored lines in the total cost graph (right) indicate 95% solution qualities for
the corresponding subsolvers. The styled vertical lines in the improvement graph (middle) and the
total cost graph (right) indicate computation times required for the corresponding learning-based
methods to reach the corresponding aforementioned solution qualities.

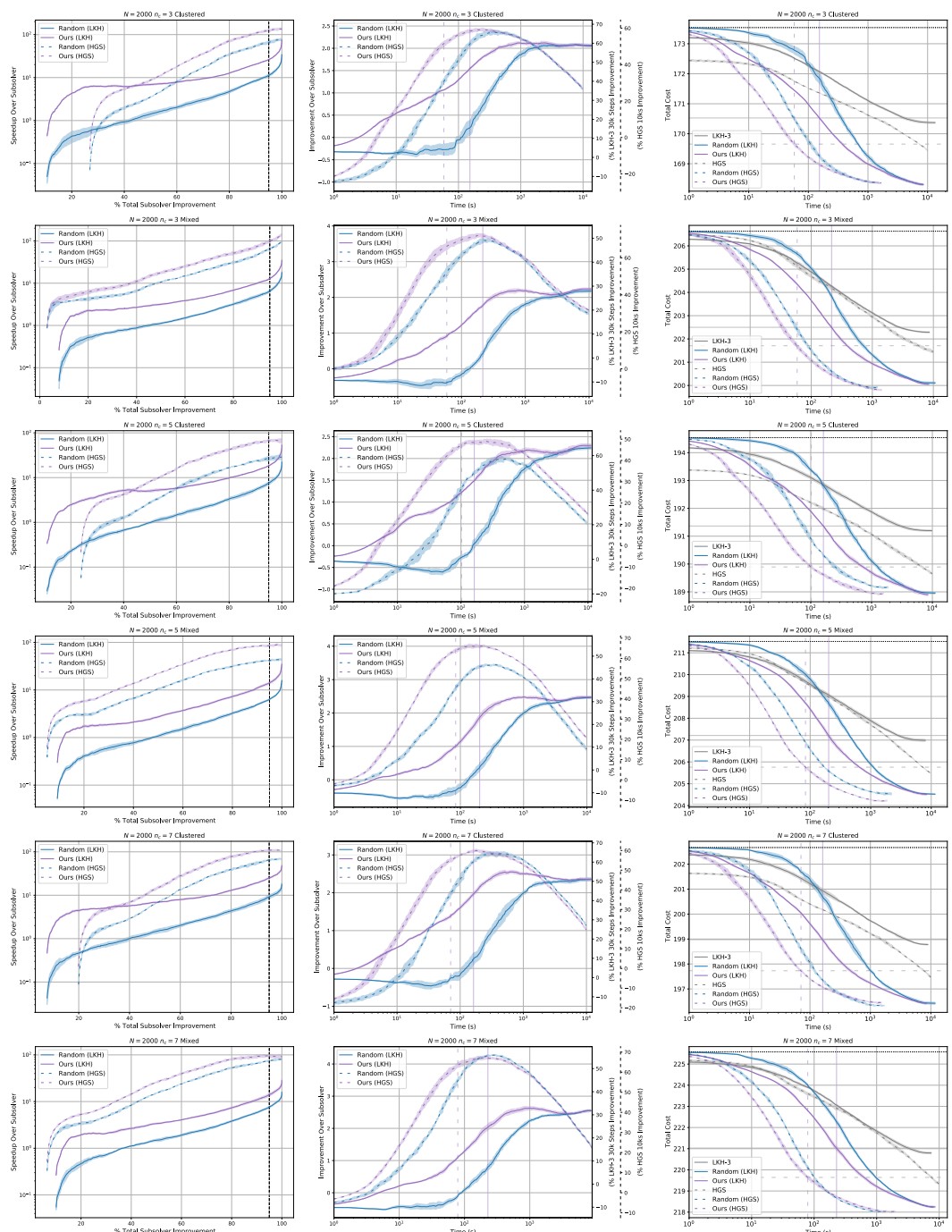

Figure 20: **Clustered and mixed CVRP distributions,** $N = 2000$**, LKH-3 and HGS subsolvers.** Graphs of speedup (left column, higher is better), improvement (middle column, higher is better), and total cost (right column, lower is better). We use loosely dashed lines to indicate HGS results. The right vertical axes in each improvement graph (middle column) indicates the improvement over the corresponding subsolver as a percentage of the best total improvement that the subsolver baseline attains. The horizontal dotted black line in the total cost graph (right) indicates the initial solution quality for all methods. The vertical dashed black line in the speedup graph (left) and the horizontal subsolver-colored lines in the total cost graph (right) indicate 95% solution qualities for the corresponding subsolvers. The styled vertical lines in the improvement graph (middle) and the total cost graph (right) indicate computation times required for the corresponding learning-based methods to reach the corresponding aforementioned solution qualities.

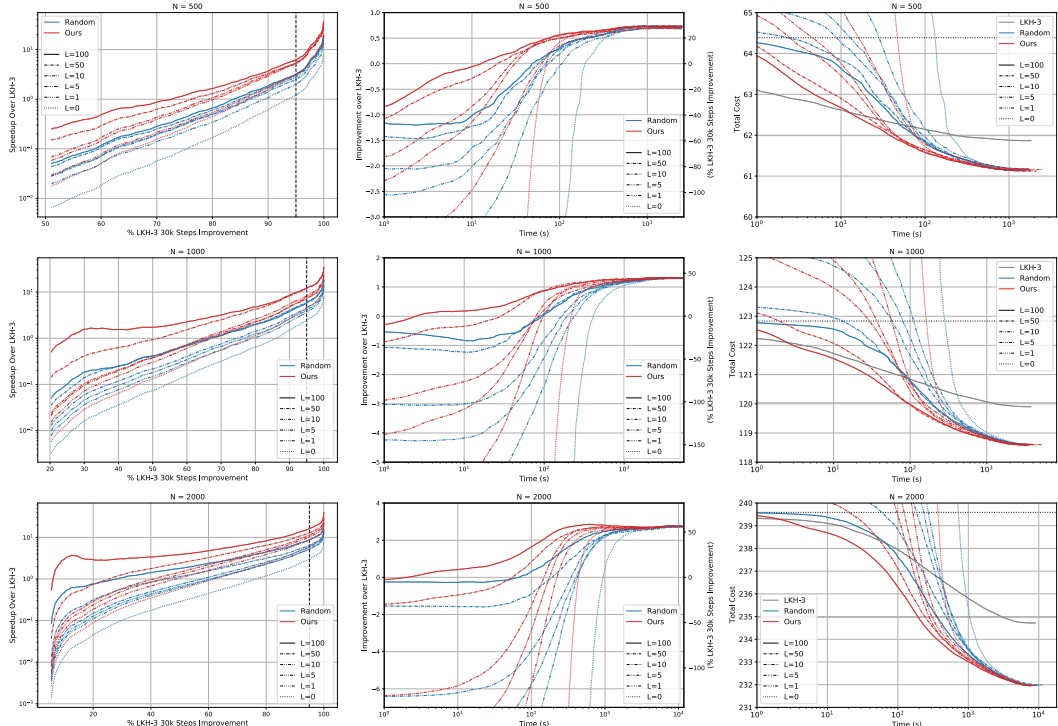

Figure 21: **Uniform CVRP distributions, various initialization qualities.** Graphs of speedup (left column, higher is better), improvement (middle column, higher is better), and total cost (right column, lower is better). The right vertical axis in each improvement graph (middle column) indicates the improvement over LKH-3 as a percentage of the total improvement that LKH-3 attains over 30k steps. We use increasingly dotted lines to indicate increasingly worse initialization methods. The horizontal dotted black line in the total cost graph (right) indicates the initial solution quality for all methods. The vertical dashed black line in the speedup graph (left) and the horizontal LKH-3-colored line in the total cost graph (right) indicate solution qualities equal to 95% of the corresponding best LKH-3 solution quality. The styled vertical lines in the improvement graph (middle) and the total cost graph (right) indicate computation times required for the corresponding learning-based methods to reach the corresponding aforementioned solution quality.

**Experimental setup.**    We describe several simpler architectures, some of which require changes to the input subproblem representation.

- **FCNN**: using the same data (positions and demands of cities) as described in Section A.1, we replace the Transformer attention layers by a fully-connected neural network.
- **Linear, MLP, RandomForest**: we design a new input representation which featurizes each subproblem into 33 summary features, including the number of cities, the bounds of the subproblem, the spread of cities, 10 radial distance percentiles of cities from the depot, 10 radial distance percentiles of cities from the subproblem centroid, and the distribution of city demands. On top of this input representation, we fit simple regression models: linear regression (Linear), a shallow fully-connected neural network (MLP), and a Random Forest regressor (RandomForest).

To train the simpler subproblem selectors, we use the same instances and training data as training the Transformer subproblem selector. Hyperparameters were lightly tuned by hand.

**Experimental results.**    The validation mean squared error for ablation architectures are similar to each other (ranging from 0.02 to 0.03) and much higher than that of our Transformer architecture (0.003). Due to the large CPU memory consumption of RandomForest at inference time and its similar validation mean squared error to the other ablation architectures, we do not evaluate RandomForest's subproblem selection ability. Figure 22 demonstrate the performance of our iterative framework with

other subproblems selectors. Interestingly, we see that ablation architectures initially, for roughly the first 5 seconds, perform comparably to our Transformer architecture. However, the quality of the solution gradually diverges from our Transformer architecture and converges to the Random baseline over the first 100 to 400 seconds. MLP, which uses summary feature inputs, performs the best out of all three ablations, though Linear offers similar performance on $N = 500$ and $1000$. Our studies show that our Transformer architecture is key for obtaining good performance over the Random baseline, especially for obtaining higher solution qualities, though summary features are surprisingly effective for subproblem selection even when combined with the interpretable Linear architecture.

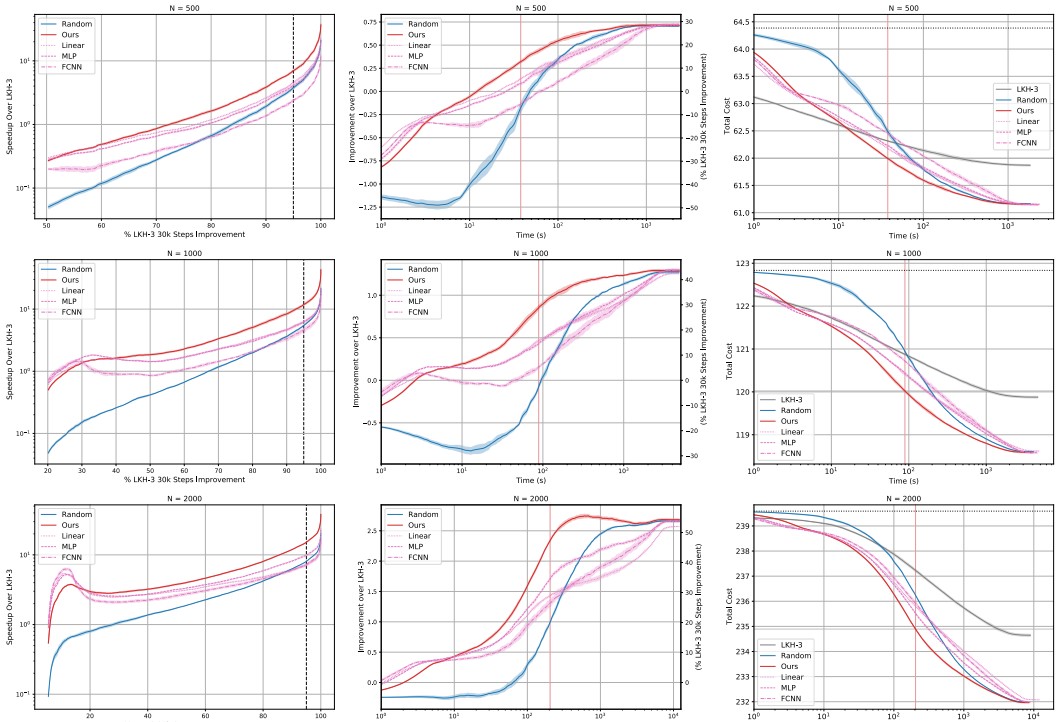

Figure 22: **Uniform CVRP distributions, ablation architectures.** Graphs of speedup (left column, higher is better), improvement (middle column, higher is better), and total cost (right column, lower is better). The right vertical axis in each improvement graph (middle column) indicates the improvement over LKH-3 as a percentage of the total improvement that LKH-3 attains over 30k steps. We indicate ablation architectures in pink. The horizontal dotted black line in the total cost graph (right) indicates the initial solution quality for all methods. The vertical dashed black line in the speedup graph (left) and the horizontal LKH-3-colored line in the total cost graph (right) indicate solution qualities equal to 95% of the corresponding best LKH-3 solution quality. The styled vertical lines in the improvement graph (middle) and the total cost graph (right) indicate computation times required for the corresponding learning-based methods to reach the corresponding aforementioned solution quality.

### A.6 Applicability of Our *learning-to-delegate* Framework to Other Problems in Combinatorial Optimization (CO)

First, our method can be seen as a learning approach to accelerate the broadly applicable POPMUSIC framework on large-scale CO problems (discussed in the related work, Section 3), where we learn principled improvement-based criteria for subproblem selection, rather than relying on random or heuristic subproblem selection. In short, as long as a restricted subproblem space can be defined, we expect an existing subsolver can be accelerated with our learning-to-delegate method. We thus give a few such example CO problems:

- **Traveling salesman problem (TSP):** in the closely related TSP problem, a restricted subproblem selection space can be defined as subpaths of a fixed length in the current solution path, as done in [41, 14]. Then, at each iteration, the solution for the unselected path segment is held fixed, while the selected subpath is finetuned by a subsolver.
- **Other graph problems:** similarly, the general POPMUSIC framework is also applied to graph problems such as map labeling (max independent set) [22], where each subproblem can be defined as the k-hop neighborhoods of a centroid node, which is also a linear subproblem selection space. Naturally, the POPMUSIC framework should also apply to related graph problems such as graph cut and minimum vertex cover.
- **Other problems with spatial or temporal locality:** similar to our subproblem space definition, problems with spatial or temporal locality can leverage this property to reduce the space. Such examples include berth allocation (which closely relates to job scheduling) [39] and p-median clustering [34].

The concrete examples given here provide a wide array of important and large-scale CO problems which we expect to benefit from learning-to-delegate. Although our method evaluation focuses on VRP, the key contribution of our method is that, given a subproblem space of tractable size, it can learn a selection strategy to drastically accelerate a subsolver, by pinpointing promising subproblems to increase the objective. On the other hand, the previous methods [41, 14, 22, 39, 34] rely on random subproblem selection or simple heuristics to select the subproblems, which expends a large amount of computation solving subproblems that already have good subsolutions.

Second, our method also has the potential to apply to CO problems with a larger subproblem space. For instance, augmenting our method with sampling-based strategies could be promising. We could consider training our subproblem regression network by **sampling** a number of subproblems at each step to generate a training set. To generate full training trajectories across multiple steps, we can 1) naively choose the greedy optimal subproblem, or 2) use an **active learning** approach to balance exploration (understanding what kinds of subproblems are promising) with exploitation (improving the objective). While 1) is a straightforward application of our current framework, 2) is an interesting extension that is worthy of further exploration.

Finally, we note that although we focus on VRPs, we design the evaluation of the approach with an eye towards generality. Specifically, VRPs have a rich family of variations, including CVRP, CVRPTW, VRPMPD, as well as different data distributions (among many others), which allows us to demonstrate a degree of generality of the method. Although they are related, each of these problem modifications considerably changes the nature of the underlying CO problem; for example, CVRPTW exhibits an element of scheduling, whereas (C)VRP does not.