# OpenReview forum: "Learning to delegate for large-scale vehicle routing"
_NeurIPS.cc/2021/Conference — NeurIPS 2021 Spotlight_

### Official Review · Reviewer_t1Zw · 2021-07-14

**Rating:** 7
**Confidence:** 3

**Summary:**

The paper introduces a local search technique for vehicle routing problems (VRP) that decomposes the general problem and delegates smaller subproblems to a subsolver, e.g. an exact VRP solver.
How to decompose the problem is learned from a generated training set using a transformer model and is the main contribution of the paper.
Experiments results with a comparison to recent works in the area show the effectiveness of the proposed method.


**Limitations And Societal Impact:**

Limitations and societal impact are adequately addressed.

**Main Review:**

It is noticed that a VRP can be decomposed into smaller subproblems that are aware of spatial locality, i.e. that a locally optimal (or at least good) subroute in an area of the overall problem is commonly part of the overall optimal (or near-optimal) route.
How to pick a good region to make improvements and find locally good routes is the main question of this research and the proposed solution is to learn a model that identifies the subproblems.

Decomposition is an important aspect for solving large combinatorial problems (such as VRP) and it is a common technique, e.g., in operations research for problem modeling. Identifying appropriate subproblems takes away a huge burden and is therefore a timely contribution in the area.
My concern is related to the generality of the method, since selecting locally connected areas in the VRP problem space is relatively easy compared to other combinatorial problems.

For the experiments presented in VRP, the same criticisms are for most (if not all) neural combinatorial optimization problems apply: The shift in training + testing effort vs. the exact solvers that only have testing effort (and there take longer), but can provide guarantees. I don't think that this problem has to be solved in this paper, but it will be a constant struggle for the time being.

My conclusion is that the presented method, when only limited to VRP, is good and interesting, but I'm missing the connection to the larger area of combinatorial optimization and solving NP-hard problems.

Questions:
- OR-Tools is selected as a baseline. It is written "We set 15 min as the time-out [..], which converges in all settings.". What is meant by converging? It is clear that OR-Tools (an exact solver) does not find the optimal solution in that time and is limited by the timeout, as shown in the table. Does converge only mean that it finds _any_ solution?
- Given my earlier criticism on generalization, what is your expectation how well other subsolvers would work in this or other problems?

Update: I read the author responses and raise my score.

**Time Spent Reviewing:**

3

---

> ### Author Response · Authors · 2021-08-11
> **Response to Reviewer t1Zw**
>
> ### In addition to reviewing our responses below, we encourage the Reviewer to take a look at our general comments, which detail several insightful ablations into our method.
>
> ## The generality of our method to other CO problems
>
> > “My concern is related to the generality of the method since selecting locally connected areas in the VRP problem space is relatively easy compared to other combinatorial problems.”
>
> Please see our general comment on “Applicability of our approach to other CO problems”. We anticipate that our framework would extend to a diverse set of CO problems, including TSP, graph problems such as max independent set, clustering, and scheduling problem.
>
> ## OR-Tools “converging”
>
> > “OR-Tools is selected as a baseline… What is meant by converging? … Does converge only mean that it finds any solution?”
>
> We acknowledge that the reported OR Tools results (Table 1) look odd due to the uniformity in taking 15 min regardless of the problem size. By “converging,” we mean that the solution stops improving **within** the allotted time (of 15 minutes), which we confirmed by also evaluating with OR Tools for longer (not reported in the paper). Since the solution quality was not among the high-performing methods, we did not investigate further to identify exactly when within the 15 minutes the solutions stopped improving. We will add this clarification in our final version.
>
> As an additional small clarification, OR-Tools is not an exact solver that tries to find an optimal solution; rather, it’s an iterative search heuristic solver that uses different local search strategies to find and improve upon already found feasible solutions. See [https://developers.google.com/optimization/routing/routing_options](https://developers.google.com/optimization/routing/routing_options). In contrast to LKH-3, however, where the number of iterations can be specified (and then the run-time varies depending on the problem size), the OR Tools uses a specified time limit, hence the uniformity of our reported time. 15 min time out hence refers to the time limit for the iterative local search procedure.
>
> ## Our framework on VRP with other solvers:
>
> > “What is your expectation of how well other subsolvers would work in this or other problems?”
>
> Fortunately, Reviewers 1 and 3 have suggested the HGS subsolver which obtains state-of-the-art performance on CVRP, and we, therefore, have integrated this subsolver into our subproblem decomposition framework, as detailed in Section “Using HGS as a subsolver”. Overall, we reached similar conclusions for the HGS subsolver as the LKH-3 subsolver: that a learned subproblem selection method may offer significant speedup over the naive subsolver or a randomized subproblem selection method.
>
> ## Conclusion
> We hope that the Reviewer found our responses and additional ablation studies insightful. If the Reviewer is satisfied with our method’s contributions and analyses, we hope that the Reviewer could increase our review score as appropriate.

---

> > ### Comment · Reviewer_t1Zw · 2021-08-26
> > **Response to authors**
> >
> > I thank the authors for their clarification.
> >
> > I suggest to include this clarification, especially regarding the convergence and using the local search variant of OR-Tools, in the manuscript.
> > Since OR-Tools also has an exact solver (which is at least in my background more common), it might be confusing otherwise.
> >
> > Given the clarifications and the responses to the other reviews  (and the expectation of the manuscript to be accordingly revised), I will raise my score from 6 to 7.

---

> > > ### Author Response · Authors · 2021-08-26
> > > **Thank you**
> > >
> > > Thank you for your suggestions! We will make sure to update the final version with the clarification on OR-Tools, as well as responses to other reviews including additional ablations, results, and discussions. Thank you again for your feedback during the rebuttal period. It truly helps us strengthen our work!

---

### Official Review · Reviewer_oqMs · 2021-07-15

**Rating:** 7
**Confidence:** 4

**Summary:**

This paper proposes an iterative framework to solve large-scale Vehicle Routing Problems. At each iteration, given the current feasible solution, a learned component selects subroutes that form a VRP subproblem. Then the subproblem is solved using a standard VRP solver (LKH-3) and the subsolution is used to update the current solution. The subproblem selector uses a Transformer encoder and predicts the subsolution cost; it is trained in a supervised way. Experimental results show that the method is able to reach the performance of the VRP solver LKH with significantly shorter computation times.


**Limitations And Societal Impact:**

Yes

**Main Review:**

Originality
---------------

The idea of this work is to decompose a large-scale VRP into a series of smaller subproblems that can be solved efficiently. This is a standard approach in Mixed Integer Linear Programming in general and for VRPs in particular (e.g. references [30] and [31] cited in the paper). The originality comes from learning the heuristic that is responsible for the subproblem definition.

The learning component itself is based on a standard Transformer encoder architecture. Using spatial information to guide the subproblems definition is also standard in decomposition techniques for the VRP (see e.g. [a, b]).

I believe the general strategy and the components are not novel. However their combination here gives good experimental results.

[a] É. Taillard, 1993, Parallel iterative search methods for vehicle routing problems.
[b] Santini et al, 2021, Decomposition strategies for vehicle routing heuristics.


Quality
----------
The paper is of good quality. I have a couple of concerns regarding the experimental validation:

In general to give a more precise sense of the improvements, I think it’s important to look at and report the optimality gaps (average and standard deviation) w.r.t the best found solution i.e. (Method cost - Best sol cost)/best sol cost. It would also be useful to indicate the size of the test set in the main text.

Since this paper proposes a decomposition framework for VRPS, I think it would be relevant to compare to other decomposition heuristics such as the cited POPMUSIC framework (or e.g. [c], the heuristics cited in [b]) that specifically target large-scale problems.
While LKH-3 is known to be an efficient heuristic to solve a variety of TSP and VRP problems, I am not sure it is the state-of-the-art solver for large scale VRPs. Do the authors have a reference that claims that?

Also the authors observe that the random problem-selector baseline achieves a similar performance within the same computation time in the uniform distribution case and with more time in the clustered mixed datasets. Do the authors have an explanation/interpretation of this observation?

In Table 1, the AM greedy gives a lower cost than the AM sampling. This is surprising as it contradicts the results reported in the original paper (admittedly for smaller problems). Do the authors have an idea of why it’s the case? What was the sample size? Since the computation time is significantly shorter than all other methods, have the authors tried increasing the sample size?

Finally, I wonder how the size of the subproblems is controlled. If the demands are very small w.r.t the vehicle capacity, the initial solution may contain only say 2 routes. Then there is only one subproblem that is the entire problem. Can this happen?

[c] Vidal, 2020, Hybrid genetic search for the cvrp: Open-source implementation and swap* neighborhood.


Clarity
---------

The paper is clear and well-written.
A suggestion: I did not get much from figures 6 and 7 whereas I would have liked to see more details in the main paper about how the proposed method was adapted to other variants of the VRP (Section 6.5).

Some typos/details:
L49-52: this is more of a description of a heuristic solution of the VRP rather than the problem itself. The vehicle capacity may return to he depot before  exhausting its capacity if it leads to a better cost.
L61: Subproblem: the depot node… —> Subproblem: CVRP in the depot node….
L14: *The* vehicle routing problem
L54: *the* depot 0.
L242: begets —> gets.


Significance
-----------------

The paper focuses on solving large-scale vehicle routing problems, using a decomposition strategy. I feel like the contribution is quite specific to VRPs. For example it's not clear how this can be adapted to the Traveling Salesman Problem (even though it is a closely related problem).

The usefulness to VRP practitioners depends on how the results of this method compare to strong decomposition heuristics for large-scale VRPs (see above Quality section).


**Time Spent Reviewing:**

7

---

> ### Author Response · Authors · 2021-08-11
> **Response to Reviewer oqMs**
>
> ### We encourage the Reviewer to look through the general comments regarding the applicability of our method to other combinatorial optimization methods, a revised version of contributions, and the additional ablation experiments which we included.
>
> ## Improvements as defined by the optimality gap:
> > “I think it’s important to look at and report the optimality gaps (average and standard deviation) w.r.t the best found solution i.e. (Method cost - Best sol cost)/best sol cost. It would also be useful to indicate the size of the test set in the main text."
>
> We thank the reviewer for these suggestions. We will make this adjustment for the figures in our final version.
>
> ## Compare to other strong VRP heuristic baselines
> > “I think it would be relevant to compare to other decomposition heuristics such as the cited POPMUSIC framework (or e.g. [c], the heuristics cited in [b])”
>
> **POPMUSIC:** the POPMUSIC framework is a general framework subject to a specific instantiation of the subproblem definition and the subsolver used for each subproblem. Typical instantiation of the POPMUSIC framework is done using the **random** subproblem selection rule [1], which is exactly one of the heuristics baselines that we chose to compare with. Indeed, our experimental results show that we obtain a significant speedup from the typical POPMUSIC random selection rule, as well as some more complex heuristics rules such as min count and max-min distance.
>
> [1] Taillard, E.D. & Voss, Stefan. (2018). “POPMUSIC”. 10.1007/978-3-319-07124-4_31.
>
> **HGS:** While the source code for [b] is unavailable, we are able to compare with the open-sourced HGS method [c], which is used in [b]. As detailed in Section “Using HGS as a subsolver”, our learned decomposition method Ours (HGS) demonstrates similar speedup over naive HGS (albeit less so for smaller subproblem sizes).
>
> [b] Santini et al, 2021, Decomposition strategies for vehicle routing heuristics.
>
> [c] Vidal, 2020, Hybrid genetic search for the cvrp: Open-source implementation and swap* neighborhood.
>
> ## LKH-3 as the SoTA solver, reference:
>
> > “While LKH-3 is known to be an efficient heuristic to solve a variety of TSP and VRP problems, I am not sure it is the state-of-the-art solver for large-scale VRPs. Do the authors have a reference that claims that?”
>
> We originally considered LKH-3 as the SoTA VRP solver based on previous learning-based VRP papers [1-2]. However, we will modify our statement on LKH-3 following the suggestion by the reviewer (and Reviewer **mEu5**). The references in [3] state that LKH-3 is the SoTA TSP solver, and we agree the extension of LKH-3 to CVRP may not lead to the SoTA CVRP solver, although it’s still quite efficient. We will also add references to the HGS solver [4-5] as one of the SoTAs, and include the additional experimental results using HGS as the subsolver. Interestingly, we see that applying HGS directly to large-scale CVRPs (N = 2000 and N = 3000) does not outperform LKH-3 by much.
>
> We note that the key point of the paper, rather than beating any specific SoTA VRP solver, is to develop a methodology to further boost **any** VRP solver’s performance to solve large problem instances. As such, we compare with LKH-3 in the paper because we use the same LKH-3 as the subsolver of our pipeline. From this fair comparison, the performance gain and speedup from our method exactly represents the contribution of our subproblem-selection network. Our findings are confirmed for the HGS subsolver as well. Given the strong experimental validation, we believe that our subproblem-selector is able to improve upon many VRP solver’s performance beyond POPMUSIC. This is particularly helpful in the large-scale VRP (as well as other large-scale CO problem) regime, where current solvers lack the ability to scale, and hence our work can help bridge the performance gap via our decomposition subproblem selector.
>
> [1] Sykora, Quinlan, Mengye Ren, and Raquel Urtasun. "Multi-agent routing value iteration network." International Conference on Machine Learning. PMLR, 2020.
>
> [2] Lu, Hao, Xingwen Zhang, and Shuang Yang. "A learning-based iterative method for solving vehicle routing problems." International Conference on Learning Representations. 2019.
>
> [3] Helsgaun, Keld. "An extension of the Lin-Kernighan-Helsgaun TSP solver for constrained traveling salesman and vehicle routing problems." Roskilde: Roskilde University (2017).
>
> [4] Vidal, T., et al. (2012). A hybrid genetic algorithm for multidepot and periodic vehicle routing problems. Operations Research, 60(3), 611-624.
>
> [5] Vidal, T. (2020). Hybrid genetic search for the CVRP: Open-source implementation and SWAP* neighborhood. arXiv:2012.10384
>
> ## Explanation / Interpretation for the random problem-selector
> > “Also the authors observe that the random problem-selector baseline achieves a similar performance within the same computation time in the uniform distribution case and with more time in the clustered mixed datasets. Do the authors have an explanation/interpretation of this observation?”
>
> We are uncertain which of our observations the reviewer is referring to and hope the reviewer can clarify further. In both the uniform and clustered CVRP distributions, our learned method offers a speedup of around 1.5x to 2x speedup over Random at the 95% maximum LKH.
>
> ## Why AM sample performs much worse than AM greedy: What was the sample size? Increase Sample Size?
> > “In Table 1, the AM greedy gives a lower cost than the AM sampling. This is surprising as it contradicts the results reported in the original paper (admittedly for smaller problems). Do the authors have an idea of why it’s the case? What was the sample size? Since the computation time is significantly shorter than all other methods, have the authors tried increasing the sample size?”
>
> We believe that AM sampling suffers from the fact that the solution space is exponentially larger with our large problem instance sizes. Therefore, any small stochasticity and imperfections from AM sampling may autoregressively compound much more in our setting than in previous AM settings.
>
> We use a sample size of 1280 for all three N’s (detailed in Supplementary Material A.2 (p17)), which is the same setup as used in the original AM paper for VRP 20, 50, and 100.
>
> We perform the additional experiment of increasing the number of samples in AM. We observe that additional samples improve the AM sampling solution a little bit (the table entries are the average and standard deviation of the distance, lower the better, on the test set). For $N=500$, AM sampling seems to be able to out-perform AM greedy with $50 \times 1280$ more samples, yet for $N=1000$ and $2000$ the gap between AM greedy and AM sampling is too large to close with even $200 \times 1280$ samples (which takes more than 1 hour/problem to generate). Additionally, for all $N$, the solution quality, even with $1280 \times 200$ samples, is still far worse than other heuristic baselines such as OR Tools.
>
> | Size  \ Num samples | 1280           | 1280 * 2       | 1280 * 10      | 1280 * 50      | 1280 * 200                         | AM greedy |
> | ------------------ | -------------- | -------------- | -------------- | -------------- | ---------------------------------- | --------- |
> | N = 500            | 69.09  ± 3.47  | 68.89 ± 3.42   | 68.63 ± 3.40   | 68.39 ± 3.37   | 68.23 ± 3.35                       | 68.58     |
> | N = 1000           | 150.88 ± 9.26  | 150.36 ± 9.25  | 149.41 ± 9.17  | 148.55 ± 9.05  | 147.92 ± 8.94                      | 142.84    |
> | N = 2000           | 355.15 ± 18.54 | 353.98 ± 18.39 | 351.58 ± 18.17 | 349.20 ± 17.94 | Skipped: computation time too long | 307.86    |
>
> ## How the subproblem size is controlled and what happens if there are two routes only
>
> > “If the demands are very small w.r.t the vehicle capacity, the initial solution may contain only say 2 routes. Then there is only one subproblem that is the entire problem. Can this happen?”
>
> As our experiments performed ablations along many axes already, we used the demand distribution from previous learning-based works [1-2] and did not vary the demand distribution as another axis. As such, our paper does not consider problems with very small demands (i.e. near-TSP problems). However, this is an interesting question that may prompt further research, since one could imagine a package delivery scenario where there are hundreds of locations to be visited by a single truck. If there were very few routes, we could envision a similar learning-to-delegate method which would consider a joint VRP and TSP subproblem decomposition such that subtour subproblems are also considered (see discussion on TSP decomposition in the general discussion). TSP subproblem decomposition is discussed in the POPMUSIC for TSP literature [3].
>
> [1] Nazari, M., Oroojlooy, A., Snyder, L., and Takac, M. “Reinforcement learning for solving the vehicle routing problem.” In Advances in Neural Information Processing Systems, volume 31 (2018).
>
> [2] Kool, W., van Hoof, H., and Welling, M. “Attention, learn to solve routing problems!” In International Conference on Learning Representations (2019).
>
> [3] Helsgaun, Keld. "Using POPMUSIC for Candidate Set Generation in the Lin-Kernighan-Helsgaun TSP Solver." Roskilde Universitet 7 (2018).
>
> ##  The generality of our method to other CO problems
>
> > “The paper focuses on solving large-scale vehicle routing problems, using a decomposition strategy. I feel like the contribution is quite specific to VRPs. For example it's not clear how this can be adapted to the Traveling Salesman Problem (even though it is a closely related problem)."
>
> We further expounded on this topic in Section “Applicability of our approach to other CO problems” of general comments.
>
> ## Conclusion
> We hope that our explanations and additional ablation results adequately addressed the reviewer’s concerns. If appropriate, we encourage the reviewer to increase our review score.

---

> ### Comment · Reviewer_oqMs · 2021-08-24
> **Update after reading authors responses**
>
> The authors have provided a high quality rebuttal, including additional experiments that strengthen their arguments.
> They have appropriately addressed my concerns, in particular I appreciated:
> * The comments about how this contribution could be applied to a variety of CO problems,
> * The application of the proposed framework to the stronger HGS solver for the CVRP,
> * The precise answers to my questions and the clarification of the main contributions,
> * The answers to relevant points raised by other reviewers.
>
> I am happy to revise my score from 5 to 7.

---

> > ### Author Response · Authors · 2021-08-24
> > **Thank you**
> >
> > Thank you for taking a careful look at our rebuttal and even our responses to other reviewers. We're happy that our rebuttal alleviated the reviewer's main concerns. The reviewer's feedbacks led to insightful experiments which improved our own confidence in our method as well. We appreciate the time that the reviewer has taken throughout the process to thoroughly review our work!

---

### Official Review · Reviewer_7sjD · 2021-07-15

**Rating:** 7
**Confidence:** 4

**Summary:**

The paper proposes a decomposition method for large-scale vehicle routing problems (VRPs) that leverage supervised learning. Specifically, the proposed methodology uses regression to identify a smaller, more tractable subproblem from a given initial solution, which is then addressed by another solver and incorporated into a full solution. Numerical results compare different state-of-the-art heuristics (e.g., LKH-3) and exact approaches (e.g., OR Tool) on different variations of the VRP with up to 3,000 nodes.


**Limitations And Societal Impact:**

No specific concerns.

**Main Review:**

**Originality**

I did find the specific setup novel and interesting. There are, however, several related routing methods that incorporate learning when decide which cities to group together to solve them separately, which is a classical idea for large-scale VRPs. For example,

Poullet, Julie. Leveraging machine learning to solve The vehicle Routing Problem with Time Windows. Diss. Massachusetts Institute of Technology, 2020.

and references therein. Perhaps the key difference is that existing methods formulate the problem as either as a more traditional classification task or as a unsupervised learning problem, while in this paper the authors use a regression/classification to learn how to "break" an existing solution, which I found quite interesting. My suggestion is for authors to better relate with that associated VRP literature, as there are strong parallels between approaches.

**Quality**

The numerical study is very thorough to the best of my analysis. All the parameters are explained and detailed clearly. I believe some details could be moved from the Appendix to the main text, or at least expanded, in particular those concerning the  training details and some comments concerning the ablation study.

My main questions are:

(a) What is the impact of the initial solution to the overall approach? It seems that the initial routes could greatly modify the possible subproblems one uses both for training as well as for the actual runs. Furthermore, are the final gaps/times obtained robust with respect to the quality of the initial solution provided?

(b) The number of subproblems is quite small. What are the gains if we relax or restrict the distance thresholds? That is, how one could determine the numbers to use? Any insights here would be extremely helpful, such as a sensitivity analysis on the solution time/quality impact. Moreover, is there a way to avoid enumerating all subproblems, possibly as a future work?

**Clarity**

The paper is very well written and easy to follow. The source code is also provided and the supplemental material is excellent.

**Significance**

In my view, the paper presents simple but effective ideas for large-scale VRPs that improve upon the state of the art heuristics under certain settings. The numerical study is thorough and insightful. I believe the technique can be of interest to a larger audience beyond routing as it is applicable to other large-scale combinatorial problems that are amenable to constructive approaches, such as the maximum stable set. I have some small concerns about the impact of the initial solution and the requirement to enumerate subproblems, but I believe they are easy to be clarified.

**Time Spent Reviewing:**

3

---

> ### Author Response · Authors · 2021-08-11
> **Response to Reviewer 7sjD (1)**
>
> ## Related literature on learning to decompose for VRP
> > “There are, however, several related routing methods that incorporate learning when decide which cities to group together to solve them separately… My suggestion is for authors to better relate with that associated VRP literature”
> Thank you for the encouraging feedback and the pointer to the work by Julie Poillet. We will include a discussion of it and other related learning works in our paper. The key differences between our work and theirs are summarized here:
>
> 1. Experimental outcomes: Whereas our method demonstrates experimental success in terms of both speedup and improvement in solution quality, we see that Poillet’s decomposition actually does not demonstrate either. Notably, Poillet’s decomposition approach performs worse than its underlying subsolver on the entire problem directly (p91).
> 2. VRP problem setup: Although both Poillet and we study VRP, the specific problem setup does have important differences. Poillet uses a less standard problem setup, where the primary objective of their VRPTW problem setup is to minimize the number of vehicles in the final solution. In contrast, our objective is to minimize the total traveling distance, which we expect to be more difficult because we must solve both embedded knapsack and a traveling salesman problems, rather than only a knapsack problem with time windows.
> 3. The role of learning in the decomposition method: Additionally, although both we and Poillet propose a decomposition method, the structure of the method and subsequently the role of learning are notably different. In Poillet, the decomposition is determined once (not iteratively) via a set covering integer program (IP), and learning is used as a proxy for the IP objective (specifically, to classify whether the subproblem has 1-5 vehicles or “other”). Our work, on the other hand, directly uses learning to iteratively select among subproblems given a decomposition (via a regression task). That is, both methods leverage both learning and classical approaches, but the role of learning within the decomposition method is distinct.
>
> ## Initial solution ablation study
>
> > “What is the impact of the initial solution to the overall approach? … Furthermore, are the final gaps/times obtained robust with respect to the quality of the initial solution provided?”
>
> We really appreciate the reviewer for asking this question, as we had the same question as well. We conducted an additional ablation which varies the quality of the initial solutions:
>
> 1. Naive: we assign customers randomly to each route until the route reaches capacity
> 2. Partitioning ($L$): we partition the problem instance into 10 angular sectors, then run LKH-3 for $L$ iterations on each cluster. Note that our paper uses Partitioning ($L = 100$), so here we additionally examine $L \in \\{1, 5, 10, 50\\}$.
>
> The following table shows the effect of the choice of initialization on the speedup for obtaining percentages (25%, 50%, 75%, 100%) of LKH-3’s total improvement. Both Ours and Random are based on the LKH-3 subsolver. We see that our method is robust to the initialization used, and that our trained method with even the worst initialization (Naive) approaches the Random baseline with the best initialization ($L = 100$). The solution quality at convergence does not seem to be affected.
>
> |         | 25% |        |      | 50% |        |      | 75% |        |      | 100% |        |       |
> | ------- | --- | ------ | ---- | --- | ------ | ---- | --- | ------ | ---- | ---- | ------ | ----- |
> |         |     | Random | Ours |     | Random | Ours |     | Random | Ours |      | Random | Ours  |
> | LKH-3   |     | 1      | 1    |     | 1      | 1    |     | 1      | 1    |      | 1      | 1     |
> | Naive   |     | 0.09   | 0.18 |     | 0.32   | 0.64 |     | 1      | 2.03 |      | 8.15   | 16.28 |
> | L = 1   |     | 0.22   | 0.37 |     | 0.69   | 1.18 |     | 1.95   | 3.33 |      | 13.91  | 22.68 |
> | L = 5   |     | 0.25   | 0.47 |     | 0.73   | 1.43 |     | 1.94   | 3.95 |      | 13.57  | 26.41 |
> | L = 10  |     | 0.28   | 0.55 |     | 0.82   | 1.63 |     | 2.2    | 4.36 |      | 15.17  | 29.32 |
> | L = 50  |     | 0.54   | 1.21 |     | 1.23   | 2.61 |     | 2.81   | 5.78 |      | 16.5   | 31.63 |
> | L = 100 |     | 1.03   | **3.04** |     | 1.91   | **4.05** |     | 3.95   | **7.43** |      | 21.39  | **40.09** |
>
>
> ## Relaxed subproblem space ablation study
>
> > “The number of subproblems is quite small. What are the gains if we relax or restrict the distance thresholds? That is, how one could determine the numbers to use?”
>
> We would like to offer a few clarifying comments before citing a few experimental results, including a new ablation. In our paper, the size of the subproblem space is exactly equal to the number of routes R in the current solution ($|\mathcal S| = R \approx \frac{N \mathbb E[\text{Demand}]}{\text{Capacity}}$), since each subproblem is constructed from a route and its k-nearest neighbors routes. Setting a different k (which acts similarly to a distance threshold around the center route) would not change the number of subproblems.
>
>
> > “What are the gains if we relax or restrict the distance thresholds?”
>
> We investigated the effect of different k (“relaxing or restricting the distance thresholds”) via an ablation in our supplementary material A.6.1, where we performed found that using the smaller k = 5 initially improves solutions faster than using the larger k = 10 but eventually converges to worse solution qualities. We believe that the choice of k could be application dependent and worthy of future studies. In Section “The subproblem size (number of routes) k” of our response to Reviewer mEu5, we further discuss the extension of learning to select a k before selecting a subproblem.
>
>
> > “That is, how one could determine the numbers to use?”
>
> Beyond specifying k, which relaxes or restricts the distance thresholds but does not change the number of subproblems considered, we could also directly change the number of subproblems. One natural way to consider a larger number of subproblems is to consider multiple $k \in \mathcal K$ values simultaneously and combine the set of subproblems for each k. In this case, we obtain $|\mathcal{K}|\times R$ subproblems with $|\mathcal{K}|$ distance threshold choices and $R$ the number of routes in the current solution.
>
>
> > … Any insights here would be extremely helpful, such as a sensitivity analysis on the solution time/quality impact.
>
> Inspired by your comment, we were also curious to understand the effect of relaxing the k-nearest neighbor subproblem space, which allows sampling $|\mathcal S| \neq R$ subproblems and investigating the sensitivity to the choice of subproblem space. We define a relaxed subproblem selection space where each of the $|\mathcal S|$ subproblem is obtained by first sampling a route $r$ uniformly from the R routes, then stochastically selecting k routes with probability inversely proportional to the distance from $r$’s centroid divided by a temperature parameter $\tau$. Therefore, increasing $\tau$ allows for more diffuse subproblems. Our experiment results indicate that the stochastic subproblem space leads to notably worse performance than the deterministic k-nearest neighborhood subproblem selection space, and the performance worsens as the temperature is increased. These findings indicate:
>
> 1. The design of subproblems has a nontrivial effect on the performance. Here, we investigated the effect of a more diffuse subproblem space. However, the sensitivity we saw motivates a potential line of work to learn a more effective subproblem space.
> 2. For now, the deterministic k-nearest neighbor subproblem space used in our paper is actually quite effective.
>
> The following table demonstrates the speedups of Random and Our methods to attain the given percentages (25%, 50%, 75%, 100%) of total LKH-3 improvements. These experiments were performed in the N = 500 setting. One note for these experiments is that, given time constraints, we did not retrain the subproblem regression model on the new distribution of stochastic subproblems.
>
> | Method | Temperature ($\tau$) | 25%  | 50%  | 75%  | 100%  |
> | ------ | ---------------------- | ---- | ---- | ---- | ----- |
> | LKH-3  |                        | **1**    | **1**    | 1    | 1     |
> | Random |                        |      |      |      |       |
> |        | 1                      | 0.07 | 0.08 | 0.28 | 4.56  |
> |        | 0.1                    | 0.15 | 0.19 | 0.74 | 14.52 |
> |        | 0 (Deterministic kNN)  | 0.17 | 0.21 | 0.92 | 17.44 |
> | Ours   |                        |      |      |      |       |
> |        | 1                      | 0.13 | 0.15 | 0.44 | 6.74  |
> |        | 0.1                    | 0.34 | 0.39 | 1.18 | 19.63 |
> |        | 0 (Deterministic kNN)  | 0.64 | 0.71 | **2.08** | **33.04** |

---

> > ### Comment · Reviewer_7sjD · 2021-08-23
> > **Feedback**
> >
> > Thank you for the new ablation studies and extensive feedback, it does address many of my questions. However, it is somewhat unclear to me how authors will incorporate the large array of changes into their new draft, as it seems that significant modifications would need to be done. Could the authors kindly clarify?

---

> > > ### Author Response · Authors · 2021-08-23
> > > **Thank you for your feedback!**
> > >
> > > We appreciate the reviewer’s concern. We provided the reviewers with the tabular results here due to the graphical limitations of OpenReview, and in the final paper we will provide corresponding figures instead. We plan to incorporate the HGS ablation studies into the main text (the final version permits one more page than the initial submission), since HGS is state-of-the-art for CVRP, and this set of ablations demonstrates that our method is not specific to the LKH-3 subsolver. We plan to add the other insightful ablations into appropriate sections of the supplementary. Overall the new ablations are in line with our previous conclusions made for LKH-3, and do not require significant changes to the rest of the main text. In addition, we will add a condensed summary of the discussions on the related works and applications to other combinatorial optimization problems into the main text.

---

> ### Author Response · Authors · 2021-08-11
> **Response to Reviewer 7sjD (2)**
>
> ## Avoid enumeration
>
> > “Moreover, is there a way to avoid enumerating all subproblems, possibly as a future work?”
>
> As our subproblem regression network fits the improvement for each subproblem, any subproblem evaluated during enumeration becomes part of our dataset. The downside of naive enumeration is that there could be few subproblems with high improvements and many subproblems with low improvements, whereas we would like our dataset to focus on subproblems with high improvement (since these are what we actually select). Therefore, a smarter but more involved way to collect data is to use active learning, which would use a partially trained subproblem regression network to guide subsequent data collection by only running the subsolver on the best subproblems predicted by subproblem regression, then training the subproblem regression network on the additionally collected data. This extension of our work would permit the collection of higher quality data without resorting to enumeration.
>
> ## Conclusion
> We again thank the reviewer for the ablation suggestions, which we also found to be insightful. In addition, we hope the Reviewer may take a look at the general comments, where we detail other insightful experiments and ablations. If the new results properly addressed the Reviewer’s concerns, we hope that the Reviewer may consider to increase our score accordingly.

---

### Official Review · Reviewer_mEu5 · 2021-07-19

**Rating:** 7
**Confidence:** 5

**Summary:**

The paper proposes new ways to learn good decomposition steps in a heuristic search algorithm for the capacitated vehicle routing problem. The learning algorithm can select good seed points for the decomposition, often permitting significant speed-ups (1.5x) over other baseline decomposition algorithms (random and other simple variations). Additional experiments are conducted to evaluate the impact of the client distribution on the performance of the proposed approach.

**Limitations And Societal Impact:**

Yes

**Main Review:**

The paper is well written and generally easy to follow. The application is interesting and the experimental results will be noteworthy for both the machine learning (ML) and operations research (OR) communities.

I suggest being careful of some affirmations and statements, which can be quite misleading. The authors have a tendency to "oversell" the contribution. The CVRP literature already contains many decomposition algorithms based on geographic partitions (see, e.g., [1] for a review). Therefore it is incorrect to mention the decomposition or "the use of spatial locality" as a novelty in the contribution statement. Instead, the contribution should be focused on the learning part, the algorithm designed for it, and the numerical experiments. Similarly, the sentence "We hypothesize that in such situations, the larger problem can be efficiently approximately solved as a sequence of smaller subproblems, which can be delegated to efficient sub-solvers." is misleading: this is a well-known fact, not an author's hypothesis.

The main contribution of the paper is the design of a learning algorithm that can select seed points during decomposition steps. However, this component is only very succinctly discussed. The authors employ a very heavy hammer (a transformer network with a dense multi-head attention mechanism) but do not provide experiments supporting the choice of this architecture. I am missing an ablation study and sensitivity analysis to deconstruct the design of this method, and a comparison with simpler regression models for this learning task.

The choice of the algorithm for the subproblems and for the experimental comparisons is not properly justified. The LKH-3 solver has never been (or even pretended to be) a state-of-the-art (SOTA) algorithm for the CVRP. The use of this method as well as OR-Tools in earlier papers on the CVRP in the ML domain (around 2017) was possibly a question of ease of integration and some herd effect (subsequent papers reproducing choices made in earlier works). Papers published at NeurIPS should follow rigorous experimental standards, and this includes i) accurately representing the application domain (the CVRP has been the subject of numerous studies in the OR domain over the last 60 years) and ii) drawing experimental comparisons with algorithms that are truly SOTA. As discussed in [1] and in a long stream of subsequent comparisons, ILS [2], HGS [3,4] and SISR [5] stand out as the current state-of-the-art for CVRP in terms of solution quality. Most notably, these methods can solve instances with around 100 customers to near-optimality within seconds. Therefore, we would expect to see experimental analyses or comparisons being built on these algorithms, especially given that source code is available for some of them.

In connection with my previous comment, the authors have used [6,7] as a principled way to generate training and test data, but they did neither report experimental results on the original instances, nor comparisons with SOTA algorithms on them. This seems necessary in such a study about new heuristic results for the CVRP. The CVRPLib website (http://vrp.atd-lab.inf.puc-rio.br/index.php/en/) gathers a lot of information on these instances, and it also includes regular updates of the best-known solutions. If the authors have found new best results during these studies, they could likewise report them.

The learning algorithm is focused on the choice of the seed points, but the number of routes $k$ is kept constant through each step of the decomposition. Yet, the number of clients per route heavily depends on the instance and application (e.g., long-haul transportation may involve only a few deliveries in each route, whereas parcel-delivery applications may involve up to 100-150 deliveries per route). Consequently, using a fixed number of routes in the decomposition may give sometimes small subproblems that are too trivial, or large subproblems that are solved inefficiently in other cases. In view of this, the parameter $k$ (or any parameter that determines the size of the subproblem, e.g., a maximum number of customers) appears to be an even more important choice than the seed point used for each decomposition. Given this critical importance, why not learning this parameter instead of expecting the user to do an extensive hyperparameter calibration to select it? This would also permit better transfer between different application contexts.

Minor comment: "exorbitant amount of a time" => Quite excessive (and dependent on the termination criterion)

Overall, this paper covers important research questions and reports very interesting experiments, but the issues related to baseline comparisons are a clear weak point that leads me to recommend a weak rejection at this stage. As suggested in this report, a more thorough experimental analysis and a better positioning in relation to the current SOTA algorithms could lead me to revise my evaluation.

[1] Laporte, G., et al.  (2014). Heuristics for the vehicle routing problem. In P. Toth & D. Vigo (Eds.), Vehicle Routing: Problems, Methods, and Applications (pp. 87-116). Society for Industrial and Applied Mathematics.

[2] Subramanian, A., Uchoa, E., & Ochi, L. S. (2013). A hybrid algorithm for a class of vehicle routing problems. Computers & Operations Research, 40(10), 2519-2531.

|3] Vidal, T., et al. (2012). A hybrid genetic algorithm for multidepot and periodic vehicle routing problems. Operations Research, 60(3), 611-624.

[4] Vidal, T. (2020). Hybrid genetic search for the CVRP: Open-source implementation and SWAP* neighborhood. arXiv:2012.10384

[5] Christiaens, J., & Vanden Berghe, G. (2020). Slack induction by string removals for vehicle routing problems. Transportation Science, 54(2), 299-564.

[6] Uchoa, E., et al. (2017). New benchmark instances for the capacitated vehicle routing problem. European Journal of Operational Research, 257(3), 845-858.

[7] Arnold, F., Gendreau, M., & Sörensen, K. (2019). Efficiently solving very large scale routing problems. Computers & Operations Research, 107(1), 32-42.

=================================
           After author's responses
=================================

The authors have carefully considered my comments and conducted extensive numerical analyses, providing (i) new ablation experiments and (ii) tests with a SOTA algorithm for the CVRP. The results provide useful information and permit to fill the gaps and weaknesses mentioned in my previous review. Considering these changes, I am pleased to revise my score and recommend accepting the revised paper at NeurIPS.


**Time Spent Reviewing:**

6

---

> ### Author Response · Authors · 2021-08-11
> **Response to Reviewer mEu5**
>
> ### We’re extremely grateful for the numerous, detailed feedbacks and suggestions that Reviewer mEu5 has made. We agree with the reviewer’s assessment of our main contributions, and here we provide additional experimental results addressing Reviewer mEu5’s suggestions.
>
> ## Architecture ablation study
>
> > “I am missing an ablation study and sensitivity analysis to deconstruct the design of this method, and a comparison with simpler regression models for this learning task.”
>
> We appreciated this suggestion as it crossed our minds too. We designed the following two ablations to illuminate the importance of using the Transformer architecture.
>
> 1. FCNN: Using the same inputs (positions and demands of the customers) as the subproblem regression Transformer model in our paper, we removed the Transformer attention layers to create a new architecture which takes in the customer features of the subproblem, applied a fully connected neural network (FCNN) on each customer’s features, aggregated over the customers by summing, then applies another FCNN to obtain a prediction for the subproblem cost.
> 2. Linear, MLP, RandomForest: Instead of using the subproblem customers as inputs for the subproblem regression model, we featurize each subproblem into 33 summary features including the size of the subproblem, the bounds of the subproblem, the spread of customers (via standard deviation), 10 radial distance percentiles of customers from the depot, 10 radial distance percentiles of customers from the subproblem centroid, and the distribution (mean, std) of the customer demands. For this ablation, we fit simple scikit-learn regression models which include linear regression, shallow MLP, and Random Forest.
>
> For both ablations above, we tuned hyperparameters to obtain reasonable performance. We see that the validation mean square errors (MSE) of all ablations (0.02 to 0.03) are significantly higher than the validation MSE of our Transformer model (around 0.003). Interestingly, when we applied these models to select subproblems, we see that the solution quality (Table below) is better than Random but worse than our Transformer model at first, then deteriorates to Random or worse. From these results, we are confident that a “heavy hammer” (the Transformer architecture) is necessary to represent and evaluate the subproblem structure.
>
> We show the speedups of each method to achieve percentages (25%, 50%, 75%, 100%) of LKH-3’s maximum improvement. Random, Linear, MLP, FCNN, and Ours all use LKH-3 as the subsolver. Note that Random here refers to our uniformly random subproblem selection baseline from before and does not refer to the RandomForest model, which is omitted due to high memory usage when selecting subproblems.
>
> | N    | Method | 25%  | 50%  | 75%  | 100%  |
> | ---- | ------ | ---- | ---- | ---- | ----- |
> | 500  |        |      |      |      |       |
> |      | LKH-3  | **1**    | 1    | 1    | 1     |
> |      | Random | 0.17 | 0.45 | 1.51 | 19.07 |
> |      | Linear | 0.6  | 0.94 | 2.26 | 21.88 |
> |      | MLP    | 0.54 | 0.87 | 2.03 | 22.57 |
> |      | FCNN   | 0.31 | 0.52 | 1.2  | 12.27 |
> |      | Ours   | 0.65 | **1.24** | **3.04** | **36.18** |
> | 1000 |        |      |      |      |       |
> |      | LKH-3  | 1    | 1    | 1    | 1     |
> |      | Random | 0.23 | 0.62 | 1.88 | 17.77 |
> |      | Linear | 1.6  | 1.57 | 3.02 | 17.59 |
> |      | MLP    | **1.74** | 1.6  | 3.07 | 18.82 |
> |      | FCNN   | 1.1  | 1.01 | 2.08 | 15.23 |
> |      | Ours   | 1.59 | **2.1**  | **4.64** | **33.86** |
> | 2000 |        |      |      |      |       |
> |      | LKH-3  | 1    | 1    | 1    | 1     |
> |      | Random | 1.03 | 1.91 | 3.95 | 21.39 |
> |      | Linear | 2.68 | 3.29 | 4.84 | 17.86 |
> |      | MLP    | 2.79 | 3.51 | 5.65 | 23.85 |
> |      | FCNN   | 2.48 | 3.12 | 4.82 | 18.68 |
> |      | Ours   | **3.04** | **4.05** | **7.43** | **40.09** |
>
>
> ## SoTA VRP algorithm and additional results with HGS
>
> > “The LKH-3 solver has never been (or even pretended to be) a state-of-the-art (SOTA) algorithm for the CVRP…  ILS [2], HGS [3,4] and SISR [5] stand out as the current state-of-the-art for CVRP in terms of solution quality”
>
> We really appreciated these references to superior methods for CVRP, which we were not previously aware of. We acknowledged that we used CVRP in part due to the herd effect. As described in the general discussion, we have followed this suggestion by adopting the open-source HGS CVRP solver as our subsolver for additional results, which we present in Section “Using HGS as a Subsolver”. Nevertheless, we would like to reiterate that one advantage of using the LKH-3 subsolver is that its open-source implementation supports numerous CVRP variants (such as CVRPTW and VRPMPD), which the open-source HGS solver does not.
>
>
> > “[The authors] did neither report experimental results on the original [Uchoa 2017 and Arnold 2019] instances, nor comparisons with SOTA algorithms on them”
>
> This is a good point, and there are several reasons why we did not initially compare with these methods. Mostly importantly, distribution of the ratio between demand and capacity (which dictates the average length of routes) varies significantly for these problems, from 3 customers per route to 100+ customers per route. In our data collection for CVRP, we fixed the demand distribution to be uniform within [1, 9] as we already vary the problem parameters along three axes: number of customers, customer distribution type (uniform, mixed, and clustered), and number of cluster centers. Our paper demonstrates that learning outperforms baselines within and somewhat out of the training distribution (as shown in the transfer learning experiments to larger problem sizes and unseen customer distributions), but we felt that it would be difficult to generalize from our training demand distribution to Uchoa 2017 and Arnold 2019’s demand distributions. The focus of our work aims to use learning to identify subproblems which may improve the fastest; we are less focused on the eventual convergence optimality derived from running the method for a long time.
>
> ## The subproblem size (number of routes in each subproblem) k
>
> > “Given this critical importance [of parameter k], why not learning this parameter instead of expecting the user to do an extensive hyperparameter calibration to select it?”
>
> We agree that the subproblem size parameter k is critically important to the solution quality and that an adaptive k should take the current problem instance and the current solution into account. However, as k is not a differentiable parameter, we would need to apply reinforcement learning and/or tree search to learn an adaptive k which optimizes an objective which factors in both solution quality and solution time. Jointly learning to select k and to select the best subproblem given k is even more difficult to optimize. One interesting direction to explore is to train one or more subproblem regression models with data from several different k then apply reinforcement learning and/or tree search to learn to select what k to use. Doing so decomposes the joint optimization into a more tractable optimization for “k selection” and our current optimization for subproblem selection.
>
> ## Modifying affirmative statements
>
> > “I suggest being careful of some affirmations and statements, which can be quite misleading.”
>
> Taking into account of the Reviewer’s feedback, we have provided a revised assessment of our  contributions in Section “Revised contributions”.
>
> ## Conclusion
> We really appreciate the Reviewer’s actionable feedbacks and we made great efforts to address with additional ablation studies. We hope that the Reviewer will take these new results into account and increase our score if we have adequately addressed the main concerns.

---

> > ### Comment · Reviewer_mEu5 · 2021-08-23
> > **Feedback**
> >
> > The authors have carefully considered my comments and conducted extensive numerical analyses, providing (i) new ablation experiments and (ii) tests with a SOTA algorithm for the CVRP. The results provide useful information and permit to fill the gaps and weaknesses mentioned in my previous review. Considering these changes, I am pleased to revise my score and recommend accepting the revised paper at NeurIPS.

---

> > > ### Author Response · Authors · 2021-08-23
> > > **Thank you**
> > >
> > > We're very thankful that the reviewer took the time to revise their assessment! We will incorporate the additional ablations, results, and discussions into the final version. Thank you again for your detailed feedbacks and suggestions throughout the whole process!

---

### Author Response · Authors · 2021-08-11
**General response to all reviewers (1)**

### **We are grateful to each of the reviewers for their time and their detailed and constructive comments, which we truly feel have contributed to strengthening the work. Here we first address general themes before answering each reviewer’s specific questions.**

## Applicability of our approach to other CO problems

Multiple reviewers (Reviewers **oqMs** and **t1Zw**) raised important points about the generality of our approach for other combinatorial optimization (CO) problems. Reviewer **oqMs** notes “I feel like the contribution is quite specific to VRPs. For example, it's not clear how this can be adapted to the Traveling Salesman Problem (even though it is a closely related problem). ” While Reviewer **t1Zw** says “I'm missing the connection to the larger area of combinatorial optimization and solving NP-hard problems.”

First, we share the perspective of Reviewer **7sjD**, who notes that our method is amenable to other CO problems such as the maximum independent set. More generally, our method can be seen as a learning approach to accelerate the broadly applicable POPMUSIC framework on large-scale CO problems (discussed in the related work section), where we learn principled improvement-based criteria for subproblem selection, rather than relying on random or heuristic subproblem selection. In short, as long as a restricted subproblem space can be defined, we expect an existing subsolver can be accelerated with our learning-to-delegate method. We thus give a few such example CO problems:

1. **Traveling salesman problem:** For example, in the closely related TSP problem, a restricted subproblem selection space can be defined as subpaths of a fixed length L in the current solution path, as done in [1-2] Then, at each iteration, the solution for the unselected path segment is held fixed, while the selected subpath is fine-tuned by a subsolver.
2. **Other graph problems:** Similarly, the general POPMUSIC framework is also applied to graph problems such as map labeling (max independent set) [3], where each subproblem can be defined as the k-hop neighborhoods of a centroid node, which is also a linear subproblem selection space. Naturally, the POPMUSIC framework should also apply to related graph problems such as graph cut and minimum vertex cover.
3. **Other problems with spatial or temporal locality:** Similar to our subproblem space definition, problems with spatial or temporal locality can leverage this property to reduce the space. Such examples include berth allocation (which closely relates to job scheduling) [4] and p-median clustering [5].

The concrete examples given here provide a wide array of important and large-scale CO problems which we expect to benefit from learning-to-delegate. Although our method evaluation focuses on VRP, the key contribution of our method is that, given a subproblem space of tractable size, it can learn a selection strategy to drastically accelerate a subsolver, by pinpointing promising subproblems to increase the objective. On the other hand, the previous methods [1-5] (and including [6]) rely on random subproblem selection or simple heuristics to select the subproblems, which can expend a large amount of computation solving subproblems that already have good solutions.

Second, while it was not the focus of the present work, our method also has the potential to apply to CO problems with a larger subproblem space. For instance, augmenting our method with sampling-based strategies could be promising. We could consider training our subproblem regression network by **sampling** a number of subproblems at each step to generate a training set. To generate full training trajectories across multiple steps, we can 1) naively choose the greedy optimal subproblem, or 2) use an **active learning** approach to balance exploration (understanding what kinds of subproblems are promising) with exploitation (improving the objective).  While 1) is a straightforward application of our current framework, 2) is an interesting extension that is worthy of further exploration.

Finally, we would like to note that although we focused on VRPs, we have designed the evaluation of the approach with an eye towards generality. Specifically, VRPs have a rich family of variations, including CVRP, CVRPTW, VRPMPD, as well as different data distributions (among many others), which allows us to demonstrate a degree of generality of the method. Although they are related, each of these problem modifications considerably changes the nature of the underlying CO problem; for example, CVRPTW exhibits an element of scheduling, whereas (C)VRP does not.

[1] Taillard, Éric D., and Keld Helsgaun. "POPMUSIC for the travelling salesman problem." European Journal of Operational Research 272.2 (2019): 420-429.

[2] Helsgaun, Keld. “Using POPMUSIC for Candidate Set Generation in the Lin-Kernighan-Helsgaun TSP Solver." Roskilde Universitet 7 (2018).

[3] Alvim, Adriana CF, and Éric D. Taillard. “POPMUSIC for the point feature label placement problem." European Journal of Operational Research 192.2 (2009): 396-413.

[4] Lalla-Ruiz, E. and Voss, S. (2016). “Popmusic as a metaheuristic for the berth allocation problem. Annals of Mathematics and Artificial Intelligence”, 76(1-2):173–189.

[5] Taillard, É. D. (2003). “Heuristic methods for large centroid clustering problems. Journal of heuristics”, 9(1):51–73.

[6] Santini et al, 2021, “Decomposition strategies for vehicle routing heuristics.”

---

### Author Response · Authors · 2021-08-11
**General response to all reviewers (2)**

##  Using HGS as a subsolver

We learned from multiple reviewers that LKH-3 is not the state-of-the-art solver for CVRP and that several solvers, including the open-source HGS solver, offer superior performance. We appreciate the feedback and have integrated HGS into our framework as a drop-in replacement for LKH-3 in the CVRP analyses for both uniform and clustered CVRP distributions. However, we note that the open-source HGS solver does not tolerate VRP variants such as CVRPTW and VRPMPD, and that one advantage of the open-source LKH-3 is its versatility for various VRPs and TSPs. Moreover, as HGS is designed for problem instances of N < 1000, its performance on large-scale N = 2000 and N = 3000 is not significantly better than LKH-3.

Here we summarize our framework’s results using HGS as the subsolver, with our previous LKH-3-based results as a comparison. Note that we keep our LKH-3 based initialization for all experiments to better compare with our previous results. In each case, we run HGS for same amount of time that we run LKH-3 previously: [2000s, 4500s, 15000s, 40000s] respectively for N = [500, 1000, 2000, 3000].  In the following tables, we report the speedup of each method for attaining percentages (25%, 50%, 75%, 100%) of total improvement of the naive HGS baseline. E.g. a speedup of 2 for attaining 25% of total improvement of the HGS baseline means that the method attains 25% of the total improvement of HGS is 2x faster than the HGS itself. N/A denotes that the method does not reach the corresponding threshold of performance.

Here we report the speedup for all methods in the N = [500, 1000, 2000, 3000] uniform CVRP distribution. We see that Ours (HGS) consistently offers a 1.3-1.5x speedup over Random (HGS) while also offering significant speedup over naive HGS, especially for large N. In one case, for N = 500, Ours (HGS) does not attain 100% total HGS improvement; this is unsurprising as HGS is designed specifically for size 100 to 1000 problem instances. We also include our previous results with LKH-3 for comparison; we see that Ours (LKH-3) still offers significant speedup over naive HGS for large instance sizes.

| N    | Method       | 25%          | 50%          | 75%           | 100%          |
| ---- | ------------ | ------------ | ------------ | ------------- | ------------- |
| 500  |              |              |              |               |               |
|      | LKH-3        | 0.37 ± 0.03  | 0.05 ± 0.01  | N/A           | N/A           |
|      | HGS          | 1.00 ± 0.01  | 1.00 ± 0.04  | 1.00 ± 0.05   | **1.00 ± 0.04**   |
|      | Random (LKH) | 0.11 ± 0.01  | 0.16 ± 0.01  | 0.16 ± 0.01   | N/A           |
|      | Ours (LKH)   | 0.34 ± 0.01  | 0.29 ± 0.01  | 0.22 ± 0.02   | N/A           |
|      | Random (HGS) | 0.68 ± 0.05  | 1.04 ± 0.05  | 1.78 ± 0.07   | N/A           |
|      | Ours (HGS)   | **1.08 ± 0.01**  | **1.60 ± 0.02**  | **2.60 ± 0.15**   | N/A           |
| 1000 |              |              |              |               |               |
|      | LKH-3        | 1.56 ± 0.08  | 0.45 ± 0.02  | N/A           | N/A           |
|      | HGS          | 1.00 ± 0.02  | 1.00 ± 0.05  | 1.00 ± 0.03   | 1.00 ± 0.03   |
|      | Random (LKH) | 0.41 ± 0.01  | 0.70 ± 0.02  | 1.10 ± 0.01   | N/A           |
|      | Ours (LKH)   | 2.54 ± 0.06  | 1.85 ± 0.05  | 1.94 ± 0.11   | N/A           |
|      | Random (HGS) | 3.23 ± 0.23  | 5.80 ± 0.23  | 11.16 ± 0.19  | **30.87 ± 1.10**  |
|      | Ours (HGS)   | **5.93 ± 0.04**  | **9.08 ± 0.12**  | **15.86 ± 0.24**  | 25.43 ± 2.05  |
| 2000 |              |              |              |               |               |
|      | LKH-3        | 1.01 ± 0.05  | 0.87 ± 0.03  | 0.65 ± 0.06   | N/A           |
|      | HGS          | 1.00 ± 0.03  | 1.00 ± 0.04  | 1.00 ± 0.04   | 1.00 ± 0.02   |
|      | Random (LKH) | 1.08 ± 0.06  | 2.13 ± 0.05  | 5.48 ± 0.07   | 8.60 ± 0.16   |
|      | Ours (LKH)   | 2.92 ± 0.04  | 4.30 ± 0.03  | 10.29 ± 0.06  | 11.71 ± 0.29  |
|      | Random (HGS) | 8.00 ± 0.15  | 16.44 ± 0.29 | 45.56 ± 0.82  | 86.21 ± 2.05  |
|      | Ours (HGS)   | **10.21 ± 0.13** | **22.93 ± 0.21** | **66.51 ± 0.48**  | **107.42 ± 1.60** |
| 3000 |              |              |              |               |               |
|      | LKH-3        | 0.65 ± 0.02  | 0.99 ± 0.03  | 0.99 ± 0.04   | N/A           |
|      | HGS          | 1.00 ± 0.02  | 1.01 ± 0.04  | 1.00 ± 0.02   | 1.00 ± 0.02   |
|      | Random (LKH) | 1.15 ± 0.03  | 3.98 ± 0.05  | 12.00 ± 0.08  | 19.50 ± 0.19  |
|      | Ours (LKH)   | 1.89 ± 0.03  | 6.08 ± 0.06  | 16.52 ± 0.27  | 19.31 ± 0.44  |
|      | Random (HGS) | 8.61 ± 0.14  | 30.15 ± 0.71 | 94.24 ± 2.02  | 169.12 ± 1.50 |
|      | Ours (HGS)   | **11.69 ± 0.56** | **43.02 ± 1.52** | **137.33 ± 4.80** | **199.68 ± 7.70** |

Here we report the speedup for all methods in the N = 2000 Clustered and Mixed CVRP distributions. We see that Ours (HGS) consistently offers a 1.5x to 2x speedup over Random (HGS) while maintaining significant speedup over naive HGS.

| num clusters | Distribution | Method       | 25%          | 50%          | 75%          | 100%           |
| --- | ------------ | ------------ | ------------ | ------------ | ------------ | -------------- |
| 3   |              |              |              |              |              |                |
|     | Clustered    |              |              |              |              |                |
|     |              | LKH-3        | 0.59 ± 0.03  | 0.33 ± 0.01  | 0.24 ± 0.07  | N/A            |
|     |              | HGS          | 1.00 ± 0.05  | 1.00 ± 0.03  | 1.00 ± 0.07  | 1.00 ± 0.03    |
|     |              | Random (LKH) | 0.55 ± 0.06  | 0.97 ± 0.10  | 4.76 ± 0.22  | 12.31 ± 0.53   |
|     |              | Ours (LKH)   | 3.69 ± 0.13  | 3.04 ± 0.05  | 10.60 ± 0.18 | 21.09 ± 0.89   |
|     |              | Random (HGS) | 3.58 ± 0.32  | 6.00 ± 0.17  | 29.75 ± 1.55 | 79.00 ± 5.70   |
|     |              | Ours (HGS)   | **10.73 ± 0.70** | **14.36 ± 0.65** | **60.25 ± 3.58** | **135.85 ± 7.84**  |
|     | Mixed        |              |              |              |              |                |
|     |              | LKH-3        | 1.29 ± 0.10  | 1.00 ± 0.03  | 0.72 ± 0.04  | N/A            |
|     |              | HGS          | 1.00 ± 0.07  | 1.00 ± 0.07  | 0.99 ± 0.05  | 1.00 ± 0.11    |
|     |              | Random (LKH) | 1.01 ± 0.06  | 1.71 ± 0.07  | 3.83 ± 0.25  | 11.40 ± 0.62   |
|     |              | Ours (LKH)   | 3.20 ± 0.07  | 4.16 ± 0.06  | 7.76 ± 0.17  | 20.10 ± 1.01   |
|     |              | Random (HGS) | 7.87 ± 0.59  | 13.28 ± 0.33 | 28.63 ± 1.76 | 96.72 ± 5.54   |
|     |              | Ours (HGS)   | **12.16 ± 1.10** | **22.07 ± 1.64** | **48.36 ± 3.79** | **141.24 ± 12.08** |
| 5   |              |              |              |              |              |                |
|     | Clustered    |              |              |              |              |                |
|     |              | LKH-3        | 0.52 ± 0.05  | 0.25 ± 0.02  | N/A          | N/A            |
|     |              | HGS          | 1.00 ± 0.01  | 1.01 ± 0.07  | 1.00 ± 0.07  | 1.00 ± 0.05    |
|     |              | Random (LKH) | 0.40 ± 0.01  | 0.76 ± 0.04  | 2.75 ± 0.14  | 6.16 ± 0.36    |
|     |              | Ours (LKH)   | 2.71 ± 0.13  | 2.30 ± 0.05  | 6.37 ± 0.20  | 9.39 ± 0.82    |
|     |              | Random (HGS) | 2.42 ± 0.28  | 4.29 ± 0.24  | 14.77 ± 0.64 | 28.99 ± 4.27   |
|     |              | Ours (HGS)   | **8.98 ± 0.70**  | **12.31 ± 0.65** | **38.50 ± 2.81** | **67.19 ± 6.68**   |
|     | Mixed        |              |              |              |              |                |
|     |              | LKH-3        | 1.33 ± 0.15  | 0.92 ± 0.07  | N/A          | N/A            |
|     |              | HGS          | 1.00 ± 0.04  | 0.99 ± 0.05  | 1.00 ± 0.04  | 1.00 ± 0.01    |
|     |              | Random (LKH) | 1.04 ± 0.08  | 2.07 ± 0.12  | 4.58 ± 0.13  | 7.07 ± 0.37    |
|     |              | Ours (LKH)   | 2.83 ± 0.06  | 4.84 ± 0.16  | 10.04 ± 0.57 | 10.65 ± 0.33   |
|     |              | Random (HGS) | 6.35 ± 0.27  | 12.83 ± 0.50 | 29.00 ± 1.05 | 43.61 ± 1.61   |
|     |              | Ours (HGS)   | **12.63 ± 0.44** | **27.17 ± 0.69** | **65.55 ± 2.19** | **88.76 ± 4.45**   |
| 7   |              |              |              |              |              |                |
|     | Clustered    |              |              |              |              |                |
|     |              | LKH-3        | 0.62 ± 0.10  | 0.41 ± 0.01  | N/A          | N/A            |
|     |              | HGS          | 1.00 ± 0.04  | 1.00 ± 0.02  | 1.00 ± 0.08  | 1.00 ± 0.02    |
|     |              | Random (LKH) | 0.60 ± 0.06  | 1.21 ± 0.09  | 4.44 ± 0.42  | 8.01 ± 0.17    |
|     |              | Ours (LKH)   | 3.55 ± 0.12  | 3.91 ± 0.08  | 11.77 ± 0.40 | 15.23 ± 0.91   |
|     |              | Random (HGS) | 4.16 ± 0.39  | 8.88 ± 0.25  | 34.38 ± 1.67 | 68.41 ± 2.86   |
|     |              | Ours (HGS)   | **10.93 ± 1.01** | **18.92 ± 1.41** | **67.79 ± 4.59** | **108.19 ± 2.90**  |
|     | Mixed        |              |              |              |              |                |
|     |              | LKH-3        | 1.00 ± 0.04  | 1.01 ± 0.03  | 0.44 ± 0.07  | N/A            |
|     |              | HGS          | 0.99 ± 0.08  | 1.00 ± 0.05  | 1.00 ± 0.07  | 1.00 ± 0.05    |
|     |              | Random (LKH) | 1.10 ± 0.03  | 2.59 ± 0.13  | 5.96 ± 0.21  | 7.84 ± 0.17    |
|     |              | Ours (LKH)   | 2.63 ± 0.09  | 5.31 ± 0.15  | 10.55 ± 0.55 | 11.78 ± 0.80   |
|     |              | Random (HGS) | 7.50 ± 0.36  | 18.70 ± 0.58 | 48.32 ± 1.87 | 80.87 ± 3.54   |
|     |              | Ours (HGS)   | **12.52 ± 0.51** | **31.84 ± 1.51** | **75.53 ± 4.10** | **92.10 ± 7.85**   |

---

### Author Response · Authors · 2021-08-11
**General response to all reviewers (3)**

## Revised contributions

Based on various reviewers’ feedback and our additional ablations, we have slightly rephrased our overall contributions here.

- We propose a learning-based algorithm called *learning-to-delegate* for selecting subproblems, which solves large-scale VRPs by iteratively identifying and solving smaller subproblems.
- Despite the high dimensionality and NP-hardness of the subproblems, we design a regression algorithm that effectively extracts shared information from graph-structured subproblems to predict their potential improvement, using a Transformer architecture as the algorithm’s backbone.
- We show through extensive validation that learning-to-delegate offers significant speedups and/or objective improvements over both its base solver and random (or heuristic) subproblem selection, for a variety of VRP variants, problem distributions, and VRP solvers.


## Additional ablation studies

To further illuminate various aspects of our method inquired by individual reviewers, we conducted additional ablation studies which may be of interest to all reviewers. Here provide a list of all additional ablation studies conducted so far and their location in the forum

1. Using the HGS subsolver instead of the LKH-3 subsolver in uniform and clustered CVRP: Section **“Using HGS as a subsolver”** in **general response to all reviewers (2)**
2. Importance of using Transformer attention compared to simpler models: Section **“Architecture ablation study”** in response to Reviewer **mEu5**
3. Sensitivity of our method to initial solution: Section **“Initial solution ablation study”** in response to Reviewer **7sjD**
4. Using k stochastic nearest neighbors with temperature $\tau$ rather than k deterministic nearest neighbors: Section **“Relaxed subproblem space ablation study”** in response to Reviewer **7sjD**

---

### Decision · Program_Chairs · 2021-09-27

**Decision:**

Accept (Spotlight)

**Comment:**

This paper proposes and analyzes a learning augmented local search algorithm to solve large-scale VRPs. Overall, all reviewers found novelty and merit in the work that was presented, but a number of questions and potential concerns of the work were posed. In their rebuttal, the author(s) were very thorough and convincing in answering these questions/concerns, and in many cases additional experiments were performed to provide additional insights in response to specific reviewer questions.

As a result of these interactions, the unanimous decision of the reviewers was to accept the paper (with multiple reviewers raising their final scores), continent on the the author(s) commitment to revise the paper to incorporate the following: (1) the revised statement of contributions they provided in the rebuttal, (2) an expanded discussion of other related learning-based approaches to this problem, (3) the additional comparative results produced relative to HGS, which was suggested as a better SOTA approach to compare against than LKH-3 and (4)  the results of new ablation experiments that were performed in response to specific reviewer queries that further enhance the paper's conclusions. In response to a direct question regarding how the authors would accomplish all revisions during the rebuttal phase, the author(s) explicitly agreed to this set of changes, and we are taking them at their word.